# Autoformer: Decomposition Transformers with Auto-Correlation for Long-Term Series Forecasting

**Haixu Wu, Jiehui Xu, Jianmin Wang, Mingsheng Long** (✉)
School of Software, BNRist, Tsinghua University, China
{whx20,xjh20}@mails.tsinghua.edu.cn, {jimwang,mingsheng}@tsinghua.edu.cn

## Abstract

Extending the forecasting time is a critical demand for real applications, such as extreme weather early warning and long-term energy consumption planning. This paper studies the *long-term forecasting* problem of time series. Prior Transformer-based models adopt various self-attention mechanisms to discover the long-range dependencies. However, intricate temporal patterns of the long-term future prohibit the model from finding reliable dependencies. Also, Transformers have to adopt the sparse versions of point-wise self-attentions for long series efficiency, resulting in the information utilization bottleneck. Going beyond Transformers, we design *Autoformer* as a novel decomposition architecture with an *Auto-Correlation* mechanism. We break with the pre-processing convention of series decomposition and renovate it as a basic inner block of deep models. This design empowers Autoformer with progressive decomposition capacities for complex time series. Further, inspired by the stochastic process theory, we design the Auto-Correlation mechanism based on the series periodicity, which conducts the dependencies discovery and representation aggregation at the sub-series level. Auto-Correlation outperforms self-attention in both efficiency and accuracy. In long-term forecasting, Autoformer yields state-of-the-art accuracy, with a 38% relative improvement on six benchmarks, covering five practical applications: energy, traffic, economics, weather and disease. Code is available at this repository: https://github.com/thuml/Autoformer.

## 1 Introduction

Time series forecasting has been widely used in energy consumption, traffic and economics planning, weather and disease propagation forecasting. In these real-world applications, one pressing demand is to extend the forecast time into the far future, which is quite meaningful for the long-term planning and early warning. Thus, in this paper, we study the *long-term forecasting* problem of time series, characterizing itself by the large length of predicted time series. Recent deep forecasting models [41, 17, 20, 28, 23, 29, 19, 35] have achieved great progress, especially the Transformer-based models. Benefiting from the self-attention mechanism, Transformers obtain great advantage in modeling long-term dependencies for sequential data, which enables more powerful big models [7, 11].

However, the forecasting task is extremely challenging under the long-term setting. First, it is unreliable to discover the temporal dependencies directly from the long-term time series because the dependencies can be obscured by entangled temporal patterns. Second, canonical Transformers with self-attention mechanisms are computationally prohibitive for long-term forecasting because of the quadratic complexity of sequence length. Previous Transformer-based forecasting models [41, 17, 20] mainly focus on improving self-attention to a *sparse* version. While performance is significantly improved, these models still utilize the point-wise representation aggregation. Thus, in the process of efficiency improvement, they will sacrifice the information utilization because of the sparse point-wise connections, resulting in a bottleneck for long-term forecasting of time series.

35th Conference on Neural Information Processing Systems (NeurIPS 2021).

To reason about the intricate temporal patterns, we try to take the idea of decomposition, which is a standard method in time series analysis [1, 27]. It can be used to process the complex time series and extract more predictable components. However, under the forecasting context, it can only be used as the *pre-processing* of past series because the future is unknown [15]. This common usage limits the capabilities of decomposition and overlooks the potential future interactions among decomposed components. Thus, we attempt to go beyond pre-processing usage of decomposition and propose a generic architecture to empower the deep forecasting models with immanent capacity of progressive decomposition. Further, decomposition can ravel out the entangled temporal patterns and highlight the inherent properties of time series [15]. Benefiting from this, we try to take advantage of the series periodicity to renovate the point-wise connection in self-attention. We observe that the sub-series at the same phase position among periods often present similar temporal processes. Thus, we try to construct a series-level connection based on the process similarity derived by series periodicity.

Based on the above motivations, we propose an original **Autoformer** in place of the Transformers for long-term time series forecasting. Autoformer still follows residual and encoder-decoder structure but renovates Transformer into a decomposition forecasting architecture. By embedding our proposed decomposition blocks as the inner operators, Autoformer can progressively separate the long-term trend information from predicted hidden variables. This design allows our model to alternately decompose and refine the intermediate results during the forecasting procedure. Inspired by the stochastic process theory [8, 24], Autoformer introduces an **Auto-Correlation** mechanism in place of self-attention, which discovers the sub-series similarity based on the series periodicity and aggregates similar sub-series from underlying periods. This series-wise mechanism achieves $\mathcal{O}(L \log L)$ complexity for length-$L$ series and breaks the information utilization bottleneck by expanding the point-wise representation aggregation to sub-series level. Autoformer achieves the state-of-the-art accuracy on six benchmarks. The contributions are summarized as follows:

- To tackle the intricate temporal patterns of the long-term future, we present *Autoformer* as a decomposition architecture and design the inner decomposition block to empower the deep forecasting model with immanent progressive decomposition capacity.

- We propose an *Auto-Correlation* mechanism with dependencies discovery and information aggregation at the series level. Our mechanism is beyond previous self-attention family and can simultaneously benefit the computation efficiency and information utilization.

- Autoformer achieves a 38% relative improvement under the long-term setting on six benchmarks, covering five real-world applications: energy, traffic, economics, weather and disease.

## 2 Related Work

### 2.1 Models for Time Series Forecasting

Due to the immense importance of time series forecasting, various models have been well developed. Many time series forecasting methods start from the classic tools [32, 9]. ARIMA [6, 5] tackles the forecasting problem by transforming the non-stationary process to stationary through differencing. The filtering method is also introduced for series forecasting [18, 10]. Besides, recurrent neural networks (RNNs) models are used to model the temporal dependencies for time series [36, 26, 40, 22]. DeepAR [28] combines autoregressive methods and RNNs to model the probabilistic distribution of future series. LSTNet [19] introduces convolutional neural networks (CNNs) with recurrent-skip connections to capture the short-term and long-term temporal patterns. Attention-based RNNs [39, 30, 31] introduce the temporal attention to explore the long-range dependencies for prediction. Also, many works based on temporal convolution networks (TCN) [34, 4, 3, 29] attempt to model the temporal causality with the causal convolution. These deep forecasting models mainly focus on the temporal relation modeling by recurrent connections, temporal attention or causal convolution.

Recently, Transformers [35, 38] based on the self-attention mechanism shows great power in sequential data, such as natural language processing [11, 7], audio processing [14] and even computer vision [12, 21]. However, applying self-attention to long-term time series forecasting is computationally prohibitive because of the quadratic complexity of sequence length $L$ in both memory and time. LogTrans [20] introduces the local convolution to Transformer and proposes the LogSparse attention to select time steps following the exponentially increasing intervals, which reduces the complexity to $\mathcal{O}(L(\log L)^2)$. Reformer [17] presents the local-sensitive hashing (LSH) attention and reduces the complexity to $\mathcal{O}(L \log L)$. Informer [41] extends Transformer with KL-divergence based ProbSparse

attention and also achieves $\mathcal{O}(L \log L)$ complexity. Note that these methods are based on the vanilla Transformer and try to improve the self-attention mechanism to a *sparse* version, which still follows the point-wise dependency and aggregation. In this paper, our proposed Auto-Correlation mechanism is based on the inherent periodicity of time series and can provide series-wise connections.

## 2.2 Decomposition of Time Series

As a standard method in time series analysis, time series decomposition [1, 27] deconstructs a time series into several components, each representing one of the underlying categories of patterns that are more predictable. It is primarily useful for exploring historical changes over time. For the forecasting tasks, decomposition is always used as the *pre-processing* of historical series before predicting future series [15, 2], such as Prophet [33] with trend-seasonality decomposition and N-BEATS [23] with basis expansion and DeepGLO [29] with matrix decomposition. However, such pre-processing is limited by the plain decomposition effect of historical series and overlooks the hierarchical interaction between the underlying patterns of series in the long-term future. This paper takes the decomposition idea from a new progressive dimension. Our Autoformer harnesses the decomposition as an inner block of deep models, which can progressively decompose the hidden series throughout the whole forecasting process, including both the past series and the predicted intermediate results.

## 3   Autoformer

The time series forecasting problem is to predict the most probable length-$O$ series in the future given the past length-$I$ series, denoting as *input-$I$-predict-$O$*. The *long-term forecasting* setting is to predict the long-term future, i.e. larger $O$. As aforementioned, we have highlighted the difficulties of long-term series forecasting: handling intricate temporal patterns and breaking the bottleneck of computation efficiency and information utilization. To tackle these two challenges, we introduce the decomposition as a builtin block to the deep forecasting model and propose *Autoformer* as a decomposition architecture. Besides, we design the *Auto-Correlation* mechanism to discover the period-based dependencies and aggregate similar sub-series from underlying periods.

## 3.1   Decomposition Architecture

We renovate Transformer [35] to a deep decomposition architecture (Figure 1), including the inner series decomposition block, Auto-Correlation mechanism, and corresponding Encoder and Decoder.

**Series decomposition block**    To learn with the complex temporal patterns in long-term forecasting context, we take the idea of decomposition [1, 27], which can separate the series into trend-cyclical and seasonal parts. These two parts reflect the long-term progression and the seasonality of the series respectively. However, directly decomposing is unrealizable for future series because the future is just unknown. To tackle this dilemma, we present a *series decomposition block* as an inner operation of Autoformer (Figure 1), which can extract the long-term stationary trend from predicted intermediate hidden variables progressively. Concretely, we adapt the moving average to smooth out periodic fluctuations and highlight the long-term trends. For length-$L$ input series $\mathcal{X} \in \mathbb{R}^{L \times d}$, the process is:

$$
\begin{aligned}
\mathcal{X}_{\mathrm{t}} &= \mathrm{AvgPool}(\mathrm{Padding}(\mathcal{X})) \\
\mathcal{X}_{\mathrm{s}} &= \mathcal{X} - \mathcal{X}_{\mathrm{t}},
\end{aligned}
\tag{1}
$$

where $\mathcal{X}_{\mathrm{s}}, \mathcal{X}_{\mathrm{t}} \in \mathbb{R}^{L \times d}$ denote the seasonal and the extracted trend-cyclical part respectively. We adopt the $\mathrm{AvgPool}(\cdot)$ for moving average with the padding operation to keep the series length unchanged. We use $\mathcal{X}_{\mathrm{s}}, \mathcal{X}_{\mathrm{t}} = \mathrm{SeriesDecomp}(\mathcal{X})$ to summarize above equations, which is a model inner block.

**Model inputs**    The inputs of encoder part are the past $I$ time steps $\mathcal{X}_{\mathrm{en}} \in \mathbb{R}^{I \times d}$. As a decomposition architecture (Figure 1), the input of Autoformer decoder contains both the seasonal part $\mathcal{X}_{\mathrm{des}} \in \mathbb{R}^{(\frac{I}{2}+O) \times d}$ and trend-cyclical part $\mathcal{X}_{\mathrm{det}} \in \mathbb{R}^{(\frac{I}{2}+O) \times d}$ to be refined. Each initialization consists of two parts: the component decomposed from the latter half of encoder's input $\mathcal{X}_{\mathrm{en}}$ with length $\frac{I}{2}$ to provide recent information, placeholders with length $O$ filled by scalars. It's formulized as follows:

$$
\begin{aligned}
\mathcal{X}_{\mathrm{ens}}, \mathcal{X}_{\mathrm{ent}} &= \mathrm{SeriesDecomp}(\mathcal{X}_{\mathrm{en}\,\frac{I}{2}:I}) \\
\mathcal{X}_{\mathrm{des}} &= \mathrm{Concat}(\mathcal{X}_{\mathrm{ens}}, \mathcal{X}_0) \\
\mathcal{X}_{\mathrm{det}} &= \mathrm{Concat}(\mathcal{X}_{\mathrm{ent}}, \mathcal{X}_{\mathrm{Mean}}),
\end{aligned}
\tag{2}
$$

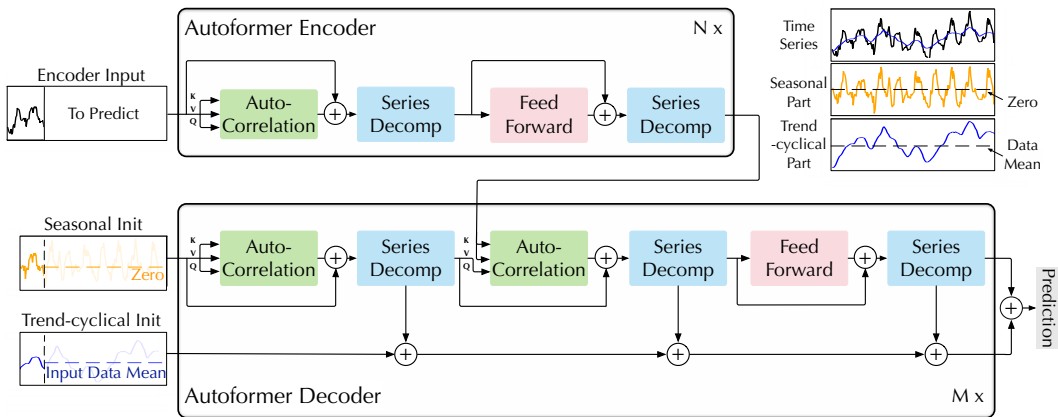

Figure 1: Autoformer architecture. The encoder eliminates the long-term trend-cyclical part by series decomposition blocks (blue blocks) and focuses on seasonal patterns modeling. The decoder accumulates the trend part extracted from hidden variables progressively. The past seasonal information from encoder is utilized by the encoder-decoder Auto-Correlation (center green block in decoder).

where $\mathcal{X}_{\mathrm{ens}}, \mathcal{X}_{\mathrm{ent}} \in \mathbb{R}^{\frac{I}{2} \times d}$ denote the seasonal and trend-cyclical parts of $\mathcal{X}_{\mathrm{en}}$ respectively, and $\mathcal{X}_0, \mathcal{X}_{\mathrm{Mean}} \in \mathbb{R}^{O \times d}$ denote the placeholders filled with zero and the mean of $\mathcal{X}_{\mathrm{en}}$ respectively.

**Encoder**  As shown in Figure 1, the encoder focuses on the seasonal part modeling. The output of the encoder contains the past seasonal information and will be used as the cross information to help the decoder refine prediction results. Suppose we have $N$ encoder layers. The overall equations for $l$-th encoder layer are summarized as $\mathcal{X}_{\mathrm{en}}^l = \mathrm{Encoder}(\mathcal{X}_{\mathrm{en}}^{l-1})$. Details are shown as follows:

$$
\begin{aligned}
\mathcal{S}_{\mathrm{en}}^{l,1}, \_ &= \mathrm{SeriesDecomp}\Big(\mathrm{Auto\text{-}Correlation}(\mathcal{X}_{\mathrm{en}}^{l-1}) + \mathcal{X}_{\mathrm{en}}^{l-1}\Big) \\
\mathcal{S}_{\mathrm{en}}^{l,2}, \_ &= \mathrm{SeriesDecomp}\Big(\mathrm{FeedForward}(\mathcal{S}_{\mathrm{en}}^{l,1}) + \mathcal{S}_{\mathrm{en}}^{l,1}\Big),
\end{aligned}
\tag{3}
$$

where "_" is the eliminated trend part. $\mathcal{X}_{\mathrm{en}}^l = \mathcal{S}_{\mathrm{en}}^{l,2}, l \in \{1, \cdots, N\}$ denotes the output of $l$-th encoder layer and $\mathcal{X}_{\mathrm{en}}^0$ is the embedded $\mathcal{X}_{\mathrm{en}}$. $\mathcal{S}_{\mathrm{en}}^{l,i}, i \in \{1, 2\}$ represents the seasonal component after the $i$-th series decomposition block in the $l$-th layer respectively. We will give detailed description of $\mathrm{Auto\text{-}Correlation}(\cdot)$ in the next section, which can seamlessly replace the self-attention.

**Decoder**  The decoder contains two parts: the accumulation structure for trend-cyclical components and the stacked Auto-Correlation mechanism for seasonal components (Figure 1). Each decoder layer contains the *inner* Auto-Correlation and *encoder-decoder* Auto-Correlation, which can refine the prediction and utilize the past seasonal information respectively. Note that the model extracts the potential trend from the intermediate hidden variables during the decoder, allowing Autoformer to progressively refine the trend prediction and eliminate interference information for period-based dependencies discovery in Auto-Correlation. Suppose there are $M$ decoder layers. With the latent variable $\mathcal{X}_{\mathrm{en}}^N$ from the encoder, the equations of $l$-th decoder layer can be summarized as $\mathcal{X}_{\mathrm{de}}^l = \mathrm{Decoder}(\mathcal{X}_{\mathrm{de}}^{l-1}, \mathcal{X}_{\mathrm{en}}^N)$. The decoder can be formalized as follows:

$$
\begin{aligned}
\mathcal{S}_{\mathrm{de}}^{l,1}, \mathcal{T}_{\mathrm{de}}^{l,1} &= \mathrm{SeriesDecomp}\Big(\mathrm{Auto\text{-}Correlation}(\mathcal{X}_{\mathrm{de}}^{l-1}) + \mathcal{X}_{\mathrm{de}}^{l-1}\Big) \\
\mathcal{S}_{\mathrm{de}}^{l,2}, \mathcal{T}_{\mathrm{de}}^{l,2} &= \mathrm{SeriesDecomp}\Big(\mathrm{Auto\text{-}Correlation}(\mathcal{S}_{\mathrm{de}}^{l,1}, \mathcal{X}_{\mathrm{en}}^N) + \mathcal{S}_{\mathrm{de}}^{l,1}\Big) \\
\mathcal{S}_{\mathrm{de}}^{l,3}, \mathcal{T}_{\mathrm{de}}^{l,3} &= \mathrm{SeriesDecomp}\Big(\mathrm{FeedForward}(\mathcal{S}_{\mathrm{de}}^{l,2}) + \mathcal{S}_{\mathrm{de}}^{l,2}\Big) \\
\mathcal{T}_{\mathrm{de}}^l &= \mathcal{T}_{\mathrm{de}}^{l-1} + \mathcal{W}_{l,1} * \mathcal{T}_{\mathrm{de}}^{l,1} + \mathcal{W}_{l,2} * \mathcal{T}_{\mathrm{de}}^{l,2} + \mathcal{W}_{l,3} * \mathcal{T}_{\mathrm{de}}^{l,3},
\end{aligned}
\tag{4}
$$

where $\mathcal{X}_{\mathrm{de}}^l = \mathcal{S}_{\mathrm{de}}^{l,3}, l \in \{1, \cdots, M\}$ denotes the output of $l$-th decoder layer. $\mathcal{X}_{\mathrm{de}}^0$ is embedded from $\mathcal{X}_{\mathrm{des}}$ for deep transform and $\mathcal{T}_{\mathrm{de}}^0 = \mathcal{X}_{\mathrm{det}}$ is for accumulation. $\mathcal{S}_{\mathrm{de}}^{l,i}, \mathcal{T}_{\mathrm{de}}^{l,i}, i \in \{1, 2, 3\}$ represent the seasonal component and trend-cyclical component after the $i$-th series decomposition block in the $l$-th layer respectively. $\mathcal{W}_{l,i}, i \in \{1, 2, 3\}$ represents the projector for the $i$-th extracted trend $\mathcal{T}_{\mathrm{de}}^{l,i}$.

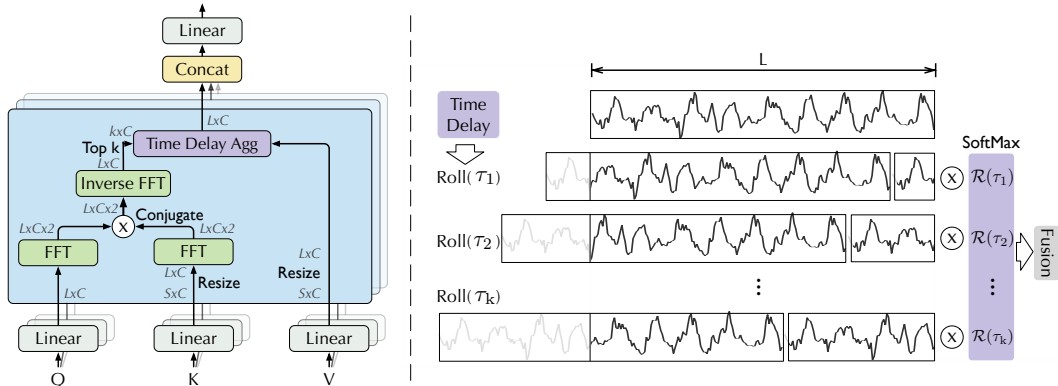

Figure 2: Auto-Correlation (left) and Time Delay Aggregation (right). We utilize the Fast Fourier Transform to calculate the autocorrelation $\mathcal{R}(\tau)$, which reflects the time-delay similarities. Then the similar sub-processes are rolled to the same index based on selected delay $\tau$ and aggregated by $\mathcal{R}(\tau)$.

The final prediction is the sum of the two refined decomposed components, as $\mathcal{W}_{\mathcal{S}} * \mathcal{X}_{\text{de}}^M + \mathcal{T}_{\text{de}}^M$, where $\mathcal{W}_{\mathcal{S}}$ is to project the deep transformed seasonal component $\mathcal{X}_{\text{de}}^M$ to the target dimension.

### 3.2 Auto-Correlation Mechanism

As shown in Figure 2, we propose the Auto-Correlation mechanism with series-wise connections to expand the information utilization. Auto-Correlation discovers the period-based dependencies by calculating the series autocorrelation and aggregates similar sub-series by time delay aggregation.

**Period-based dependencies**  It is observed that the same phase position among periods naturally provides similar sub-processes. Inspired by the stochastic process theory [8, 24], for a real discrete-time process $\{\mathcal{X}_t\}$, we can obtain the autocorrelation $\mathcal{R}_{\mathcal{X}\mathcal{X}}(\tau)$ by the following equations:

$$\mathcal{R}_{\mathcal{X}\mathcal{X}}(\tau) = \lim_{L \to \infty} \frac{1}{L} \sum_{t=1}^{L} \mathcal{X}_t \mathcal{X}_{t-\tau}. \tag{5}$$

$\mathcal{R}_{\mathcal{X}\mathcal{X}}(\tau)$ reflects the time-delay similarity between $\{\mathcal{X}_t\}$ and its $\tau$ lag series $\{\mathcal{X}_{t-\tau}\}$. As shown in Figure 2, we use the autocorrelation $\mathcal{R}(\tau)$ as the unnormalized confidence of estimated period length $\tau$. Then, we choose the most possible $k$ period lengths $\tau_1, \cdots, \tau_k$. The period-based dependencies are derived by the above estimated periods and can be weighted by the corresponding autocorrelation.

**Time delay aggregation**  The period-based dependencies connect the sub-series among estimated periods. Thus, we present the *time delay aggregation* block (Figure 2), which can roll the series based on selected time delay $\tau_1, \cdots, \tau_k$. This operation can align similar sub-series that are at the same phase position of estimated periods, which is different from the point-wise dot-product aggregation in self-attention family. Finally, we aggregate the sub-series by softmax normalized confidences.

For the single head situation and time series $\mathcal{X}$ with length-$L$, after the projector, we get query $\mathcal{Q}$, key $\mathcal{K}$ and value $\mathcal{V}$. Thus, it can replace self-attention seamlessly. The Auto-Correlation mechanism is:

$$\begin{aligned}
\tau_1, \cdots, \tau_k &= \underset{\tau \in \{1, \cdots, L\}}{\arg \operatorname{Topk}} \left( \mathcal{R}_{\mathcal{Q},\mathcal{K}}(\tau) \right) \\
\widehat{\mathcal{R}}_{\mathcal{Q},\mathcal{K}}(\tau_1), \cdots, \widehat{\mathcal{R}}_{\mathcal{Q},\mathcal{K}}(\tau_k) &= \operatorname{SoftMax}\left( \mathcal{R}_{\mathcal{Q},\mathcal{K}}(\tau_1), \cdots, \mathcal{R}_{\mathcal{Q},\mathcal{K}}(\tau_k) \right) \\
\operatorname{Auto-Correlation}(\mathcal{Q}, \mathcal{K}, \mathcal{V}) &= \sum_{i=1}^{k} \operatorname{Roll}(\mathcal{V}, \tau_i) \widehat{\mathcal{R}}_{\mathcal{Q},\mathcal{K}}(\tau_i),
\end{aligned} \tag{6}$$

where $\arg \operatorname{Topk}(\cdot)$ is to get the arguments of the Topk autocorrelations and let $k = \lfloor c \times \log L \rfloor$, $c$ is a hyper-parameter. $\mathcal{R}_{\mathcal{Q},\mathcal{K}}$ is autocorrelation between series $\mathcal{Q}$ and $\mathcal{K}$. $\operatorname{Roll}(\mathcal{X}, \tau)$ represents the operation to $\mathcal{X}$ with time delay $\tau$, during which elements that are shifted beyond the first position are re-introduced at the last position. For the encoder-decoder Auto-Correlation (Figure 1), $\mathcal{K}, \mathcal{V}$ are from the encoder $\mathcal{X}_{\text{en}}^N$ and will be resized to length-$O$, $\mathcal{Q}$ is from the previous block of the decoder.

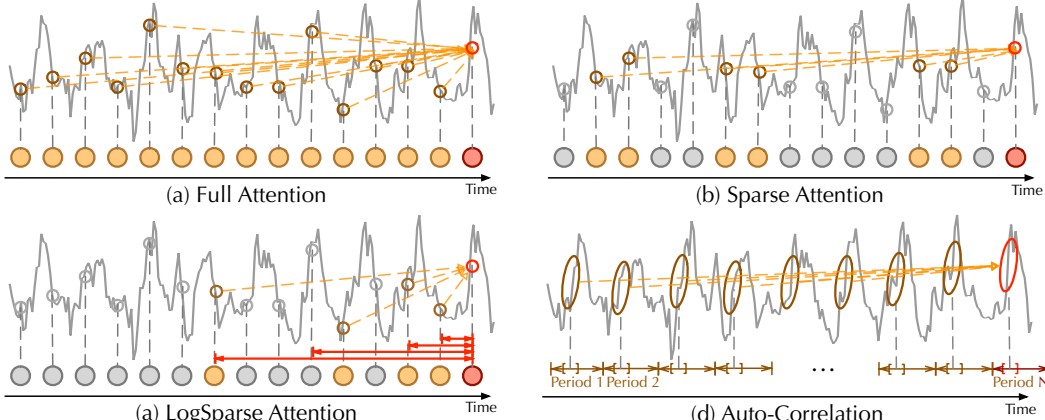

Figure 3: Auto-Correlation vs. self-attention family. Full Attention [35] (a) adapts the fully connection among all time points. Sparse Attention [17, 41] (b) selects points based on the proposed similarity metrics. LogSparse Attention [20] (c) chooses points following the exponentially increasing intervals. Auto-Correlation (d) focuses on the connections of sub-series among underlying periods.

For the multi-head version used in Autoformer, with hidden variables of $d_{\text{model}}$ channels, $h$ heads, the query, key and value for $i$-th head are $\mathcal{Q}_i, \mathcal{K}_i, \mathcal{V}_i \in \mathbb{R}^{L \times \frac{d_{\text{model}}}{h}}, i \in \{1, \cdots, h\}$. The process is:

$$\text{MultiHead}(\mathcal{Q}, \mathcal{K}, \mathcal{V}) = \mathcal{W}_{\text{output}} * \text{Concat}(\text{head}_1, \cdots, \text{head}_h)$$
$$\text{where head}_i = \text{Auto-Correlation}(\mathcal{Q}_i, \mathcal{K}_i, \mathcal{V}_i). \tag{7}$$

**Efficient computation**   For period-based dependencies, these dependencies point to sub-processes at the same phase position of underlying periods and are inherently sparse. Here, we select the most possible delays to avoid picking the opposite phases. Because we aggregate $\mathcal{O}(\log L)$ series whose length is $L$, the complexity of Equations 6 and 7 is $\mathcal{O}(L \log L)$. For the autocorrelation computation (Equation 5), given time series $\{\mathcal{X}_t\}$, $\mathcal{R}_{\mathcal{X}\mathcal{X}}(\tau)$ can be calculated by Fast Fourier Transforms (FFT) based on the Wiener–Khinchin theorem [37]:

$$\mathcal{S}_{\mathcal{X}\mathcal{X}}(f) = \mathcal{F}(\mathcal{X}_t)\mathcal{F}^*(\mathcal{X}_t) = \int_{-\infty}^{\infty} \mathcal{X}_t e^{-i2\pi tf} \mathrm{d}t \overline{\int_{-\infty}^{\infty} \mathcal{X}_t e^{-i2\pi tf} \mathrm{d}t}$$
$$\mathcal{R}_{\mathcal{X}\mathcal{X}}(\tau) = \mathcal{F}^{-1}(\mathcal{S}_{\mathcal{X}\mathcal{X}}(f)) = \int_{-\infty}^{\infty} \mathcal{S}_{\mathcal{X}\mathcal{X}}(f) e^{i2\pi f\tau} \mathrm{d}f, \tag{8}$$

where $\tau \in \{1, \cdots, L\}$, $\mathcal{F}$ denotes the FFT and $\mathcal{F}^{-1}$ is its inverse. $*$ denotes the conjugate operation and $\mathcal{S}_{\mathcal{X}\mathcal{X}}(f)$ is in the frequency domain. Note that the series autocorrelation of all lags in $\{1, \cdots, L\}$ can be calculated at once by FFT. Thus, Auto-Correlation achieves the $\mathcal{O}(L \log L)$ complexity.

**Auto-Correlation vs. self-attention family**   Different from the point-wise self-attention family, Auto-Correlation presents the series-wise connections (Figure 3). Concretely, for the temporal dependencies, we find the dependencies among sub-series based on the periodicity. In contrast, the self-attention family only calculates the relation between scattered points. Though some self-attentions [20, 41] consider the local information, they only utilize this to help point-wise dependencies discovery. For the information aggregation, we adopt the time delay block to aggregate the similar sub-series from underlying periods. In contrast, self-attentions aggregate the selected points by dot-product. Benefiting from the inherent sparsity and sub-series-level representation aggregation, Auto-Correlation can simultaneously benefit the computation efficiency and information utilization.

## 4   Experiments

We extensively evaluate the proposed Autoformer on six real-world benchmarks, covering five mainstream time series forecasting applications: energy, traffic, economics, weather and disease.

**Datasets**   Here is a description of the six experiment datasets: (1) *ETT* [41] dataset contains the data collected from electricity transformers, including load and oil temperature that are recorded every

Table 1: Multivariate results with different prediction lengths $O \in \{96, 192, 336, 720\}$. We set the input length $I$ as 36 for ILI and 96 for the others. A lower MSE or MAE indicates a better prediction.

| Models | **Autoformer** | | Informer[41] | | LogTrans[20] | | Reformer[17] | | LSTNet[19] | | LSTM[13] | | TCN[3] | |
|---|---|---|---|---|---|---|---|---|---|---|---|---|---|---|
| Metric | MSE | MAE | MSE | MAE | MSE | MAE | MSE | MAE | MSE | MAE | MSE | MAE | MSE | MAE |
| ETT* 96 | **0.255** | **0.339** | 0.365 | 0.453 | 0.768 | 0.642 | 0.658 | 0.619 | 3.142 | 1.365 | 2.041 | 1.073 | 3.041 | 1.330 |
| ETT* 192 | **0.281** | **0.340** | 0.533 | 0.563 | 0.989 | 0.757 | 1.078 | 0.827 | 3.154 | 1.369 | 2.249 | 1.112 | 3.072 | 1.339 |
| ETT* 336 | **0.339** | **0.372** | 1.363 | 0.887 | 1.334 | 0.872 | 1.549 | 0.972 | 3.160 | 1.369 | 2.568 | 1.238 | 3.105 | 1.348 |
| ETT* 720 | **0.422** | **0.419** | 3.379 | 1.388 | 3.048 | 1.328 | 2.631 | 1.242 | 3.171 | 1.368 | 2.720 | 1.287 | 3.135 | 1.354 |
| Electricity 96 | **0.201** | **0.317** | 0.274 | 0.368 | 0.258 | 0.357 | 0.312 | 0.402 | 0.680 | 0.645 | 0.375 | 0.437 | 0.985 | 0.813 |
| Electricity 192 | **0.222** | **0.334** | 0.296 | 0.386 | 0.266 | 0.368 | 0.348 | 0.433 | 0.725 | 0.676 | 0.442 | 0.473 | 0.996 | 0.821 |
| Electricity 336 | **0.231** | **0.338** | 0.300 | 0.394 | 0.280 | 0.380 | 0.350 | 0.433 | 0.828 | 0.727 | 0.439 | 0.473 | 1.000 | 0.824 |
| Electricity 720 | **0.254** | **0.361** | 0.373 | 0.439 | 0.283 | 0.376 | 0.340 | 0.420 | 0.957 | 0.811 | 0.980 | 0.814 | 1.438 | 0.784 |
| Exchange 96 | **0.197** | **0.323** | 0.847 | 0.752 | 0.968 | 0.812 | 1.065 | 0.829 | 1.551 | 1.058 | 1.453 | 1.049 | 3.004 | 1.432 |
| Exchange 192 | **0.300** | **0.369** | 1.204 | 0.895 | 1.040 | 0.851 | 1.188 | 0.906 | 1.477 | 1.028 | 1.846 | 1.179 | 3.048 | 1.444 |
| Exchange 336 | **0.509** | **0.524** | 1.672 | 1.036 | 1.659 | 1.081 | 1.357 | 0.976 | 1.507 | 1.031 | 2.136 | 1.231 | 3.113 | 1.459 |
| Exchange 720 | **1.447** | **0.941** | 2.478 | 1.310 | 1.941 | 1.127 | 1.510 | 1.016 | 2.285 | 1.243 | 2.984 | 1.427 | 3.150 | 1.458 |
| Traffic 96 | **0.613** | **0.388** | 0.719 | 0.391 | 0.684 | 0.384 | 0.732 | 0.423 | 1.107 | 0.685 | 0.843 | 0.453 | 1.438 | 0.784 |
| Traffic 192 | **0.616** | **0.382** | 0.696 | 0.379 | 0.685 | 0.390 | 0.733 | 0.420 | 1.157 | 0.706 | 0.847 | 0.453 | 1.463 | 0.794 |
| Traffic 336 | **0.622** | **0.337** | 0.777 | 0.420 | 0.733 | 0.408 | 0.742 | 0.420 | 1.216 | 0.730 | 0.853 | 0.455 | 1.479 | 0.799 |
| Traffic 720 | **0.660** | **0.408** | 0.864 | 0.472 | 0.717 | 0.396 | 0.755 | 0.423 | 1.481 | 0.805 | 1.500 | 0.805 | 1.499 | 0.804 |
| Weather 96 | **0.266** | **0.336** | 0.300 | 0.384 | 0.458 | 0.490 | 0.689 | 0.596 | 0.594 | 0.587 | 0.369 | 0.406 | 0.615 | 0.589 |
| Weather 192 | **0.307** | **0.367** | 0.598 | 0.544 | 0.658 | 0.589 | 0.752 | 0.638 | 0.560 | 0.565 | 0.416 | 0.435 | 0.629 | 0.600 |
| Weather 336 | **0.359** | **0.395** | 0.578 | 0.523 | 0.797 | 0.652 | 0.639 | 0.596 | 0.597 | 0.587 | 0.455 | 0.454 | 0.639 | 0.608 |
| Weather 720 | **0.419** | **0.428** | 1.059 | 0.741 | 0.869 | 0.675 | 1.130 | 0.792 | 0.618 | 0.599 | 0.535 | 0.520 | 0.639 | 0.610 |
| ILI 24 | **3.483** | **1.287** | 5.764 | 1.677 | 4.480 | 1.444 | 4.400 | 1.382 | 6.026 | 1.770 | 5.914 | 1.734 | 6.624 | 1.830 |
| ILI 36 | **3.103** | **1.148** | 4.755 | 1.467 | 4.799 | 1.467 | 4.783 | 1.448 | 5.340 | 1.668 | 6.631 | 1.845 | 6.858 | 1.879 |
| ILI 48 | **2.669** | **1.085** | 4.763 | 1.469 | 4.800 | 1.468 | 4.832 | 1.465 | 6.080 | 1.787 | 6.736 | 1.857 | 6.968 | 1.892 |
| ILI 60 | **2.770** | **1.125** | 5.264 | 1.564 | 5.278 | 1.560 | 4.882 | 1.483 | 5.548 | 1.720 | 6.870 | 1.879 | 7.127 | 1.918 |

* *ETT* means the ETTm2. See supplementary materials for the **full benchmark** of ETTh1, ETTh2, ETTm1.

15 minutes between July 2016 and July 2018. (2) *Electricity*[1] dataset contains the hourly electricity consumption of 321 customers from 2012 to 2014. (3) *Exchange* [19] records the daily exchange rates of eight different countries ranging from 1990 to 2016. (4) *Traffic*[2] is a collection of hourly data from California Department of Transportation, which describes the road occupancy rates measured by different sensors on San Francisco Bay area freeways. (5) *Weather*[3] is recorded every 10 minutes for 2020 whole year, which contains 21 meteorological indicators, such as air temperature, humidity, etc. (6) *ILI*[4] includes the weekly recorded influenza-like illness (ILI) patients data from Centers for Disease Control and Prevention of the United States between 2002 and 2021, which describes the ratio of patients seen with ILI and the total number of the patients. We follow standard protocol and split all datasets into training, validation and test set in chronological order by the ratio of 6:2:2 for the ETT dataset and 7:1:2 for the other datasets.

**Implementation details**   Our method is trained with L2 loss, using the ADAM [16] optimizer with an initial learning rate of $10^{-4}$. Batch size is set to 32. The training process is early stopped within 10 epochs. All experiments are repeated three times, implemented in PyTorch [25] and conducted on a single NVIDIA TITAN RTX 24GB GPUs. The hyper-parameter $c$ of Auto-Correlation is in the range of 1 to 3 to trade off performance and efficiency. See supplementary materials for standard deviations and sensitivity analysis. Autoformer contains 2 encoder layers and 1 decoder layer.

**Baselines**   We include 10 baseline methods. For the *multivariate* setting, we select three latest state-of-the-art transformer-based models: Informer [41], Reformer [17], LogTrans [20], two RNN-based models: LSTNet [19], LSTM [13] and CNN-based TCN [3] as baselines. For the *univariate* setting, we include more competitive baselines: N-BEATS[23], DeepAR [28], Prophet [33] and ARMIA [1].

[1] https://archive.ics.uci.edu/ml/datasets/ElectricityLoadDiagrams20112014
[2] http://pems.dot.ca.gov
[3] https://www.bgc-jena.mpg.de/wetter/
[4] https://gis.cdc.gov/grasp/fluview/fluportaldashboard.html

Table 2: Univariate results with different prediction lengths $O \in \{96, 192, 336, 720\}$ on typical datasets. We set the input length $I$ as 96. A lower MSE or MAE indicates a better prediction.

| Models | Autoformer | | N-BEATS[23] | | Informer[41] | | LogTrans[20] | | Reformer[17] | | DeepAR[28] | | Prophet[33] | | ARIMA[1] | |
|---|---|---|---|---|---|---|---|---|---|---|---|---|---|---|---|---|
| Metric | MSE | MAE | MSE | MAE | MSE | MAE | MSE | MAE | MSE | MAE | MSE | MAE | MSE | MAE | MSE | MAE |
| ETT 96 | **0.065** | **0.189** | 0.082 | 0.219 | 0.088 | 0.225 | 0.082 | 0.217 | 0.131 | 0.288 | 0.099 | 0.237 | 0.287 | 0.456 | 0.211 | 0.362 |
| ETT 192 | **0.118** | **0.256** | 0.120 | 0.268 | 0.132 | 0.283 | 0.133 | 0.284 | 0.186 | 0.354 | 0.154 | 0.310 | 0.312 | 0.483 | 0.261 | 0.406 |
| ETT 336 | **0.154** | **0.305** | 0.226 | 0.370 | 0.180 | 0.336 | 0.201 | 0.361 | 0.220 | 0.381 | 0.277 | 0.428 | 0.331 | 0.474 | 0.317 | 0.448 |
| ETT 720 | **0.182** | **0.335** | 0.188 | 0.338 | 0.300 | 0.435 | 0.268 | 0.407 | 0.267 | 0.430 | 0.332 | 0.468 | 0.534 | 0.593 | 0.366 | 0.487 |
| Exchange 96 | 0.241 | 0.387 | 0.156 | 0.299 | 0.591 | 0.615 | 0.279 | 0.441 | 1.327 | 0.944 | 0.417 | 0.515 | 0.828 | 0.762 | **0.112** | **0.245** |
| Exchange 192 | **0.273** | **0.403** | 0.669 | 0.665 | 1.183 | 0.912 | 1.950 | 1.048 | 1.258 | 0.924 | 0.813 | 0.735 | 0.909 | 0.974 | 0.304 | 0.404 |
| Exchange 336 | **0.508** | **0.539** | 0.611 | 0.605 | 1.367 | 0.984 | 2.438 | 1.262 | 2.179 | 1.296 | 1.331 | 0.962 | 1.304 | 0.988 | 0.736 | 0.598 |
| Exchange 720 | **0.991** | **0.768** | 1.111 | 0.860 | 1.872 | 1.072 | 2.010 | 1.247 | 1.280 | 0.953 | 1.894 | 1.181 | 3.238 | 1.566 | 1.871 | 0.935 |

## 4.1 Main Results

To compare performances under different future horizons, we fix the input length and evaluate models with a wide range of prediction lengths: 96, 192, 336, 720. This setting precisely meets the definition of long-term forecasting. Here are results on both the multivariate and univariate settings.

**Multivariate results**   As for the multivariate setting, Autoformer achieves the consistent state-of-the-art performance in all benchmarks and all prediction length settings (Table 1). Especially, under the input-96-predict-336 setting, compared to previous state-of-the-art results, Autoformer gives **74%** (1.334→0.339) MSE reduction in ETT, **18%** (0.280→0.231) in Electricity, **61%** (1.357→0.509) in Exchange, **15%** (0.733→0.622) in Traffic and **21%** (0.455→0.359) in Weather. For the input-36-predict-60 setting of ILI, Autoformer makes **43%** (4.882→2.770) MSE reduction. Overall, Autoformer yields a **38%** averaged MSE reduction among above settings. Note that Autoformer still provides remarkable improvements in the *Exchange* dataset that is **without obvious periodicity**. See supplementary materials for detailed showcases. Besides, we can also find that the performance of Autoformer changes quite steadily as the prediction length $O$ increases. It means that Autoformer retains better **long-term robustness**, which is meaningful for real-world practical applications, such as weather early warning and long-term energy consumption planning.

**Univariate results**   We list the univariate results of two typical datasets in Table 2. Under the comparison with extensive baselines, our Autoformer still achieves state-of-the-art performance for the long-term forecasting tasks. In particular, for the input-96-predict-336 setting, our model achieves **14%** (0.180→0.145) MSE reduction on the ETT dataset with obvious periodicity. For the Exchange dataset without obvious periodicity, Autoformer surpasses other baselines by **17%** (0.611→0.508) and shows greater long-term forecasting capacity. Also, we find that ARIMA [1] performs best in the input-96-predict-96 setting of the Exchange dataset but fails in the long-term setting. This situation of ARIMA can be benefited from its inherent capacity for non-stationary economic data but is limited by the intricate temporal patterns of real-world series.

## 4.2 Ablation studies

Table 3: Ablation of decomposition in multivariate ETT with MSE metric. **Ours** adopts our progressive architecture into other models. **Sep** employs two models to forecast pre-decomposed seasonal and trend-cyclical components separately. *Promotion* is the MSE reduction compared to **Origin**.

| Input-96 | Transformer[35] | | | Informer[41] | | | LogTrans[17] | | | Reformer[20] | | | Promotion | |
|---|---|---|---|---|---|---|---|---|---|---|---|---|---|---|
| Predict-O | Origin | Sep | Ours | Origin | Sep | Ours | Origin | Sep | Ours | Origin | Sep | Ours | Sep | Ours |
| 96 | 0.604 | 0.311 | **0.204** | 0.365 | 0.490 | **0.354** | 0.768 | 0.862 | **0.231** | 0.658 | 0.445 | **0.218** | 0.069 | 0.347 |
| 192 | 1.060 | 0.760 | **0.266** | 0.533 | 0.658 | **0.432** | 0.989 | 0.533 | **0.378** | 1.078 | 0.510 | **0.336** | 0.300 | 0.562 |
| 336 | 1.413 | 0.665 | **0.375** | 1.363 | 1.469 | **0.481** | 1.334 | 0.762 | **0.362** | 1.549 | 1.028 | **0.366** | 0.434 | 1.019 |
| 720 | 2.672 | 3.200 | **0.537** | 3.379 | 2.766 | **0.822** | 3.048 | 2.601 | **0.539** | 2.631 | 2.845 | **0.502** | 0.079 | 2.332 |

**Decomposition architecture**   With our proposed progressive decomposition architecture, other models can gain consistent promotion, especially as the prediction length $O$ increases (Table 3). This

verifies that our method can generalize to other models and release the capacity of other dependencies learning mechanisms, alleviate the distraction caused by intricate patterns. Besides, our architecture outperforms the pre-processing, although the latter employs a bigger model and more parameters. Especially, pre-decomposing may even bring negative effect because it neglects the interaction of components during long-term future, such as Transformer [35] predict-720, Informer [41] predict-336.

**Auto-Correlation vs. self-attention family**  As shown in Table 4, our proposed Auto-Correlation achieves the best performance under various input-$I$-predict-$O$ settings, which verifies the effectiveness of series-wise connections comparing to point-wise self-attentions (Figure 3). Furthermore, we can also observe that Auto-Correlation is memory efficiency from the last column of Table 4, which can be used in long sequence forecasting, such as input-336-predict-1440.

Table 4: Comparison of Auto-Correlation and self-attention in the multivariate ETT. We **replace** the Auto-Correlation in Autoformer with different self-attentions. The "-" indicates the out-of-memory.

| Input Length $I$ | | 96 | | | 192 | | | 336 | | |
|---|---|---|---|---|---|---|---|---|---|---|
| Prediction Length $O$ | | 336 | 720 | 1440 | 336 | 720 | 1440 | 336 | 720 | 1440 |
| Auto- | MSE | **0.339** | **0.422** | **0.555** | **0.355** | **0.429** | **0.503** | **0.361** | **0.425** | **0.574** |
| Correlation | MAE | **0.372** | **0.419** | **0.496** | **0.392** | **0.430** | **0.484** | **0.406** | **0.440** | **0.534** |
| Full | MSE | 0.375 | 0.537 | 0.667 | 0.450 | 0.554 | - | 0.501 | 0.647 | - |
| Attention[35] | MAE | 0.425 | 0.502 | 0.589 | 0.470 | 0.533 | - | 0.485 | 0.491 | - |
| LogSparse | MSE | 0.362 | 0.539 | 0.582 | 0.420 | 0.552 | 0.958 | 0.474 | 0.601 | - |
| Attention[20] | MAE | 0.413 | 0.522 | 0.529 | 0.450 | 0.513 | 0.736 | 0.474 | 0.524 | - |
| LSH | MSE | 0.366 | 0.502 | 0.663 | 0.407 | 0.636 | 1.069 | 0.442 | 0.615 | - |
| Attention[17] | MAE | 0.404 | 0.475 | 0.567 | 0.421 | 0.571 | 0.756 | 0.476 | 0.532 | - |
| ProbSparse | MSE | 0.481 | 0.822 | 0.715 | 0.404 | 1.148 | 0.732 | 0.417 | 0.631 | 1.133 |
| Attention[41] | MAE | 0.472 | 0.559 | 0.586 | 0.425 | 0.654 | 0.602 | 0.434 | 0.528 | 0.691 |

## 4.3  Model Analysis

**Time series decomposition**  As shown in Figure 4, without our series decomposition block, the forecasting model cannot capture the increasing trend and peaks of the seasonal part. By adding the series decomposition blocks, Autoformer can aggregate and refine the trend-cyclical part from series progressively. This design also facilitates the learning of the seasonal part, especially the peaks and troughs. This verifies the necessity of our proposed progressive decomposition architecture.

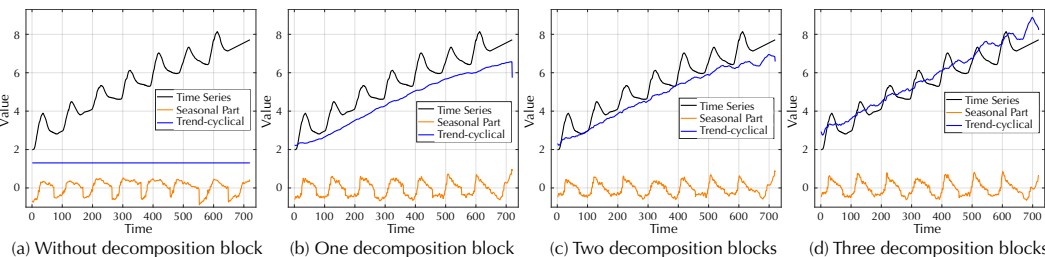

(a) Without decomposition block  (b) One decomposition block  (c) Two decomposition blocks  (d) Three decomposition blocks

Figure 4: Visualization of learned seasonal $\mathcal{X}_{\mathrm{de}}^{M}$ and trend-cyclical $\mathcal{T}_{\mathrm{de}}^{M}$ of the last decoder layer. We gradually add the decomposition blocks in decoder from left to right. This case is from ETT dataset under input-96-predict-720 setting. For clearness, we add the linear growth to raw data additionally.

**Dependencies learning**  The marked time delay sizes in Figure 5(a) indicate the most likely periods. Our learned periodicity can guide the model to aggregate the sub-series from the same or neighbor phase of periods by $\mathrm{Roll}(\mathcal{X}, \tau_i), \; i \in \{1, \cdots, 6\}$. For the last time step (declining stage), Auto-Correlation fully utilizes all similar sub-series without omissions or errors compared to self-attentions. This verifies that Autoformer can discover the relevant information more sufficiently and precisely.

**Complex seasonality modeling**  As shown in Figure 6, the lags that Autoformer learns from deep representations can indicate the real seasonality of raw series. For example, the learned lags of the

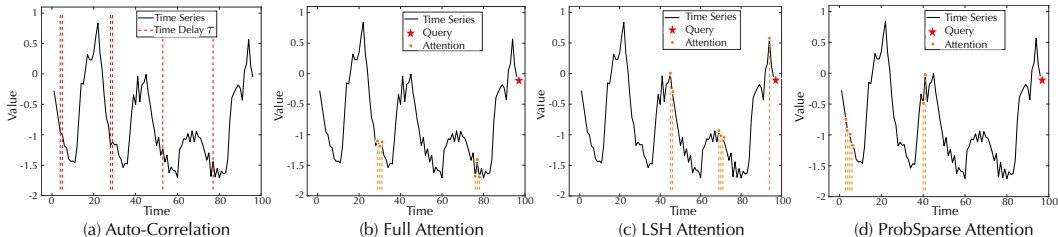

Figure 5: Visualization of learned dependencies. For clearness, we select the top-6 time delay sizes $\tau_1, \cdots, \tau_6$ of Auto-Correlation and mark them in raw series (red lines). For self-attentions, top-6 similar points with respect to the last time step (red stars) are also marked by orange points.

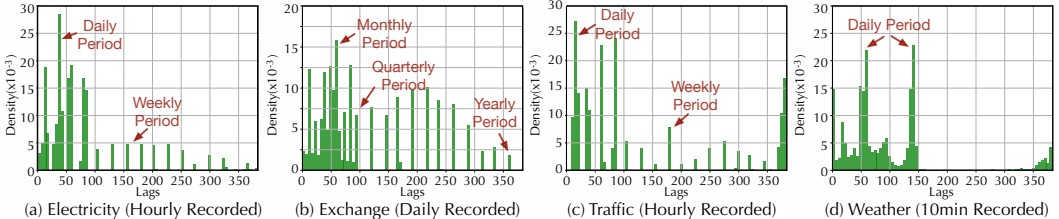

Figure 6: Statistics of learned lags. For each time series in the test set, we count the top 10 lags learned by decoder for the input-96-predict-336 task. Figure (a)-(d) are the density histograms.

daily recorded Exchange dataset present the monthly, quarterly and yearly periods (Figure 6 (b)). For the hourly recorded Traffic dataset (Figure 6 (c)), the learned lags show the intervals as 24-hours and 168-hours, which match the daily and weekly periods of real-world scenarios. These results show that Autoformer can capture the complex seasonalities of real-world series from deep representations and further provide a human-interpretable prediction.

**Efficiency analysis** We compare the running memory and time among Auto-Correlation-based and self-attention-based models (Figure 7) during the training phase. The proposed Autoformer shows $\mathcal{O}(L \log L)$ complexity in both memory and time and achieves better long-term sequences efficiency.

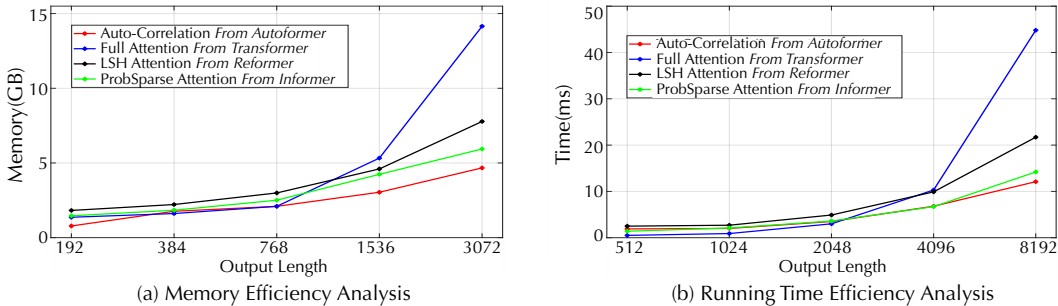

Figure 7: Efficiency Analysis. For memory, we replace Auto-Correlation with self-attention family in Autoformer and record the memory with input 96. For running time, we run the Auto-Correlation or self-attentions $10^3$ times to get the execution time per step. The output length increases exponentially.

## 5 Conclusions

This paper studies the long-term forecasting problem of time series, which is a pressing demand for real-world applications. However, the intricate temporal patterns prevent the model from learning reliable dependencies. We propose the Autoformer as a decomposition architecture by embedding the series decomposition block as an inner operator, which can progressively aggregate the long-term trend part from intermediate prediction. Besides, we design an efficient Auto-Correlation mechanism to conduct dependencies discovery and information aggregation at the series level, which contrasts clearly from the previous self-attention family. Autoformer can naturally achieve $\mathcal{O}(L \log L)$ complexity and yield consistent state-of-the-art performance in extensive real-world datasets.

## Acknowledgments and Disclosure of Funding

This work was supported by the National Natural Science Foundation of China under Grants 62022050 and 62021002, Beijing Nova Program under Grant Z201100006820041, China's Ministry of Industry and Information Technology, the MOE Innovation Plan and the BNRist Innovation Fund.

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
