# Supplementary Materials: *Autoformer: Decomposition Transformers with Auto-Correlation for Long-term Series Forecasting*

**Haixu Wu, Jiehui Xu, Jianmin Wang, Mingsheng Long (✉)**
School of Software, BNRist, Tsinghua University, China
{whx20,xjh20}@mails.tsinghua.edu.cn, {jimwang,mingsheng}@tsinghua.edu.cn

## 1 Full Benchmark on the ETT Datasets

As shown in Table 1, we build the benchmark on the four ETT datasets [14], which includes the hourly recorded ETTh1 and ETTh2, 15-minutely recorded ETTm1 and ETTm2.

Autoformer achieves sharp improvement over the state-of-the-art on various forecasting horizons. For the input-96-predict-336 long-term setting, Autoformer surpasses previous best results by **55%** (1.128→0.505) in ETTh1, **80%** (2.544→0.471) in ETTh2. For the input-96-predict-288 long-term setting, Autoformer achieves **40%** (1.056→0.634) MSE reduction in ETTm1 and **66%** (0.969→0.342) in ETTm2. These results show a **60%** average MSE reduction over previous state-of-the-art.

Table 1: Multivariate results on the four ETT datasets with predicted length as {24, 48, 168, 288, 336, 672, 720}. We fix the input length of Autoformer as 96. The experiments of the main text are on the ETTm2 dataset.

| Models | | **Autoformer** | | Informer [14] | | LogTrans [9] | | Reformer [7] | | LSTNet [8] | | LSTMa [1] | |
|---|---|---|---|---|---|---|---|---|---|---|---|---|---|
| Metric | | MSE | MAE | MSE | MAE | MSE | MAE | MSE | MAE | MSE | MAE | MSE | MAE |
| ETTh1 | 24 | **0.384** | **0.425** | 0.577 | 0.549 | 0.686 | 0.604 | 0.991 | 0.754 | 1.293 | 0.901 | 0.650 | 0.624 |
| | 48 | **0.392** | **0.419** | 0.685 | 0.625 | 0.766 | 0.757 | 1.313 | 0.906 | 1.456 | 0.960 | 0.702 | 0.675 |
| | 168 | **0.490** | **0.481** | 0.931 | 0.752 | 1.002 | 0.846 | 1.824 | 1.138 | 1.997 | 1.214 | 1.212 | 0.867 |
| | 336 | **0.505** | **0.484** | 1.128 | 0.873 | 1.362 | 0.952 | 2.117 | 1.280 | 2.655 | 1.369 | 1.424 | 0.994 |
| | 720 | **0.498** | **0.500** | 1.215 | 0.896 | 1.397 | 1.291 | 2.415 | 1.520 | 2.143 | 1.380 | 1.960 | 1.322 |
| ETTh2 | 24 | **0.261** | **0.341** | 0.720 | 0.665 | 0.828 | 0.750 | 1.531 | 1.613 | 2.742 | 1.457 | 1.143 | 0.813 |
| | 48 | **0.312** | **0.373** | 1.457 | 1.001 | 1.806 | 1.034 | 1.871 | 1.735 | 3.567 | 1.687 | 1.671 | 1.221 |
| | 168 | **0.457** | **0.455** | 3.489 | 1.515 | 4.070 | 1.681 | 4.660 | 1.846 | 3.242 | 2.513 | 4.117 | 1.674 |
| | 336 | **0.471** | **0.475** | 2.723 | 1.340 | 3.875 | 1.763 | 4.028 | 1.688 | 2.544 | 2.591 | 3.434 | 1.549 |
| | 720 | **0.474** | **0.484** | 3.467 | 1.473 | 3.913 | 1.552 | 5.381 | 2.015 | 4.625 | 3.709 | 3.963 | 1.788 |
| ETTm1 | 24 | 0.383 | 0.403 | **0.323** | **0.369** | 0.419 | 0.412 | 0.724 | 0.607 | 1.968 | 1.170 | 0.621 | 0.629 |
| | 48 | **0.454** | **0.453** | 0.494 | 0.503 | 0.507 | 0.583 | 1.098 | 0.777 | 1.999 | 1.215 | 1.392 | 0.939 |
| | 96 | **0.481** | **0.463** | 0.678 | 0.614 | 0.768 | 0.792 | 1.433 | 0.945 | 2.762 | 1.542 | 1.339 | 0.913 |
| | 288 | **0.634** | **0.528** | 1.056 | 0.786 | 1.462 | 1.320 | 1.820 | 1.094 | 1.257 | 2.076 | 1.740 | 1.124 |
| | 672 | **0.606** | **0.542** | 1.192 | 0.926 | 1.669 | 1.461 | 2.187 | 1.232 | 1.917 | 2.941 | 2.736 | 1.555 |
| ETTm2 | 24 | **0.153** | **0.261** | 0.173 | 0.301 | 0.211 | 0.332 | 0.333 | 0.429 | 1.101 | 0.831 | 0.580 | 0.572 |
| | 48 | **0.178** | **0.280** | 0.303 | 0.409 | 0.427 | 0.487 | 0.558 | 0.571 | 2.619 | 1.393 | 0.747 | 0.630 |
| | 96 | **0.255** | **0.339** | 0.365 | 0.453 | 0.768 | 0.642 | 0.658 | 0.619 | 3.142 | 1.365 | 2.041 | 1.073 |
| | 288 | **0.342** | **0.378** | 1.047 | 0.804 | 1.090 | 0.806 | 2.441 | 1.190 | 2.856 | 1.329 | 0.969 | 0.742 |
| | 672 | **0.434** | **0.430** | 3.126 | 1.302 | 2.397 | 1.214 | 3.090 | 1.328 | 3.409 | 1.420 | 2.541 | 1.239 |

35th Conference on Neural Information Processing Systems (NeurIPS 2021).

## 2 Hyper-Parameter Sensitivity

As shown in Table 2, we can verify the model robustness with respect to hyper-parameter $c$ (Equation 6 in the underline{main text}). To trade-off performance and efficiency, we set $c$ to the range of 1 to 3. It is also observed that datasets with obvious periodicity tend to have a large factor $c$, such as the ETT and Traffic datasets. For the ILI dataset without obvious periodicity, the larger factor may bring noises.

Table 2: Autoformer performance under different choices of hyper-parameter $c$ in the Auto-Correlation mechanism. We adopt the forecasting setting as input-36-predict-48 for the ILI dataset and input-96-predict-336 for the other datasets.

| Dataset | ETT | | Electricity | | Exchange | | Traffic | | Weather | | ILI | |
|---|---|---|---|---|---|---|---|---|---|---|---|---|
| Metric | MSE | MAE | MSE | MAE | MSE | MAE | MSE | MAE | MSE | MAE | MSE | MAE |
| $c=1$ | 0.339 | 0.372 | 0.252 | 0.356 | 0.511 | 0.528 | 0.706 | 0.488 | **0.348** | **0.388** | 2.754 | 1.088 |
| $c=2$ | 0.363 | 0.389 | **0.224** | **0.332** | 0.511 | 0.528 | 0.673 | 0.418 | 0.358 | 0.390 | **2.641** | **1.072** |
| $c=3$ | 0.339 | 0.372 | 0.231 | 0.338 | **0.509** | **0.524** | 0.619 | 0.385 | 0.359 | 0.395 | 2.669 | 1.085 |
| $c=4$ | **0.336** | **0.369** | 0.232 | 0.341 | 0.513 | 0.527 | **0.607** | **0.378** | 0.349 | 0.388 | 3.041 | 1.178 |
| $c=5$ | 0.410 | 0.415 | 0.273 | 0.371 | 0.517 | 0.527 | 0.618 | 0.379 | 0.366 | 0.399 | 3.076 | 1.172 |

## 3 Model Input Selection

### 3.1 Input Length Selection

Because the forecasting horizon is always fixed upon the application's demand, we need to tune the input length in real-world applications. Our study shows that the relationship between input length and model performance is dataset-specific, so we need to select the model input based on the data characteristics. For example, for the ETT dataset with obvious periodicity, an input with length-96 is enough to provide enough information. But for the ILI dataset without obvious periodicity, the model needs longer inputs to discover more informative temporal dependencies.

Table 3: Autoformer performance under different input lengths. We fix the forecasting horizon as 48 for ILI and 336 for the others. The input lengths $I$ of the ILI dataset are in the $\{24, 36, 48, 60\}$. And for the ETT and Exchange datasets, the input lengths $I$ are in the $\{96, 192, 336, 720\}$.

| Dataset | ETT | | Electricity | | Dataset | ILI | |
|---|---|---|---|---|---|---|---|
| Metric | MSE | MAE | MSE | MAE | Metric | MSE | MAE |
| $I=96$ | **0.339** | **0.372** | 0.231 | 0.338 | $I=24$ | 3.406 | 1.247 |
| $I=192$ | 0.355 | 0.392 | **0.200** | **0.316** | $I=36$ | 2.669 | 1.085 |
| $I=336$ | 0.361 | 0.406 | 0.225 | 0.335 | $I=48$ | **2.656** | **1.075** |
| $I=720$ | 0.419 | 0.430 | 0.226 | 0.346 | $I=60$ | 2.779 | 1.091 |

### 3.2 Past Information Utilization

For the decoder input of Autoformer, we attach the length-$\frac{I}{2}$ past information to the placeholder. This design is to provide recent past information to the decoder. As shown in Table 4, the model with more past information will obtain a better performance, but it also causes a larger memory cost. Thus, we set the decoder input as $\frac{I}{2} + O$ to trade off both the performance and efficiency.

Table 4: Autoformer performance under different lengths of input of the decoder. $O$, $\frac{I}{2} + O$, $I + O$ corresponds to the decoder input without past information, with half past information, with full past information respectively. We fix the forecasting setting as input-96-predict-336 on the ETT dataset.

| Decoder input length | $O$ (without past) | $\frac{I}{2} + O$ (with half past) | $I + O$ (with full past) |
|---|---|---|---|
| MSE | 0.360 | 0.339 | **0.333** |
| MAE | 0.383 | 0.372 | **0.369** |
| Memory Cost | **3029 MB** | 3271 MB | 3599 MB |

# 4 Ablation of Decomposition Architecture

In this section, we attempt to further verify the effectiveness of our proposed *progressive decomposition architecture*. We adopt more well-established decomposition algorithms as the pre-processing for separate prediction settings. As shown in Table 5, our proposed progressive decomposition architecture consistently outperforms the separate prediction (especially the long-term forecasting setting), despite the latter being with mature decomposition algorithms and twice bigger model.

Table 5: Ablation of *decomposition architecture* in ETT dataset under the input-96-predict-$O$ setting, where $O \in \{96, 192, 336, 720\}$. The backbone of separate prediction is canonical Transformer [12]. We adopt various decomposition algorithms as the pre-processing and use two Transformers to separately forecast the seasonal and trend-cyclical parts. The result is the sum of two parts prediction.

| Decomposition | Predict $O$ | 96 | | 192 | | 336 | | 720 | |
|---|---|---|---|---|---|---|---|---|---|
| | Metric | MSE | MAE | MSE | MAE | MSE | MAE | MSE | MAE |
| Separately | STL [10] | 0.523 | 0.516 | 0.638 | 0.605 | 1.004 | 0.794 | 3.678 | 1.462 |
| | Hodrick-Prescott Filter [5] | 0.464 | 0.495 | 0.816 | 0.733 | 0.814 | 0.722 | 2.181 | 1.173 |
| | Christiano-Fitzgerald Filter [2] | 0.373 | 0.458 | 0.819 | 0.668 | 1.083 | 0.835 | 2.462 | 1.189 |
| | Baxter-King Filter [13] | 0.440 | 0.514 | 0.623 | 0.626 | 0.861 | 0.741 | 2.150 | 1.175 |
| Progressively | Autoformer | **0.255** | **0.339** | **0.281** | **0.340** | **0.339** | **0.372** | **0.422** | **0.419** |

# 5 Supplementary of Main Results

## 5.1 Multivariate Showcases

To evaluate the prediction of different models, we plot the last dimension of forecasting results that are from the *test set of ETT dataset* for qualitative comparison (Figures 1, 2, 3, and 4). Our model gives the best performance among different models. Moreover, we observe that Autoformer can accurately predict the periodicity and long-term variation.

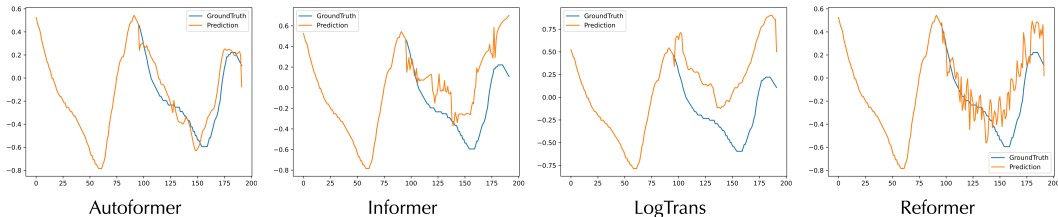

Figure 1: Prediction cases from the ETT dataset under the input-96-predict-96 setting. Blue lines are the ground truth and orange lines are the model prediction. The first part with length 96 is the input.

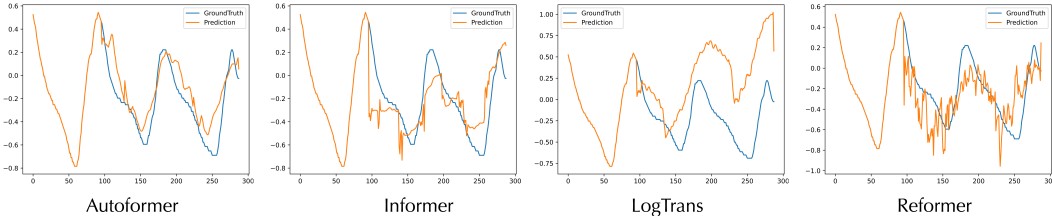

Figure 2: Prediction cases from the ETT dataset under the input-96-predict-192 setting.

## 5.2 Performance on Data without Obvious Periodicity

Autoformer yields the best performance among six datasets, even in the Exchange dataset that does not have obvious periodicity. This section will give some showcases from the test set of multivariate

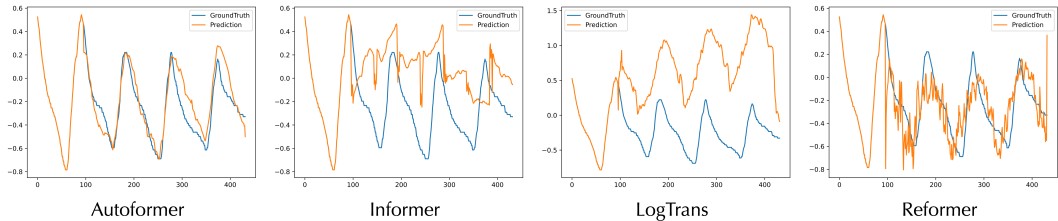

Figure 3: Prediction cases from the ETT dataset under the input-96-predict-336 setting.

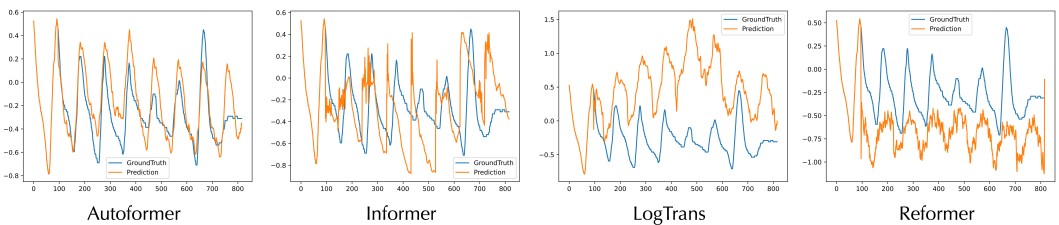

Figure 4: Prediction cases from the ETT dataset under the input-96-predict-720 setting.

**Exchange dataset for qualitative evaluation.** We observed that the series in the Exchange dataset show rapid fluctuations. And because of the inherent properties of economic data, the series does not present obvious periodicity. This aperiodicity causes extreme difficulties for prediction. As shown in Figure 5, compared to other models, Autoformer can still predict the exact long-term variations. It is verified the robustness of our model performance among various data characteristics.

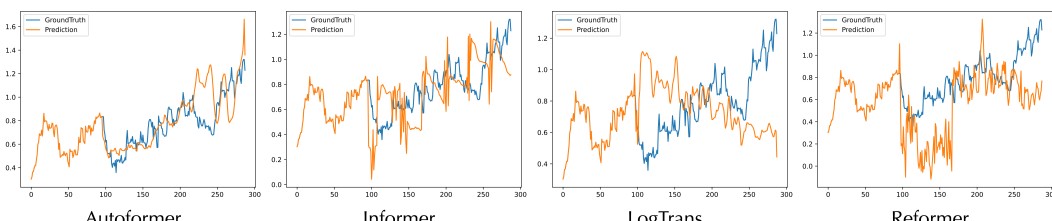

Figure 5: Prediction cases from the Exchange dataset under the input-96-predict-192 setting.

### 5.3 Univariate Forecasting Showcases

As shown in Figure 6, Autoformer gives the most accurate prediction. Compared to Informer [14], Autoformer can precisely capture the periods of the future horizon. Besides, our model provides better prediction in the center area than LogTrans [9]. Compared with Reformer [7], our prediction series is smooth and closer to ground truth. Also, the fluctuation of DeepAR [11] prediction is getting smaller as prediction length increases and suffers from the over-smoothing problem, which does not happen in our Autoformer.

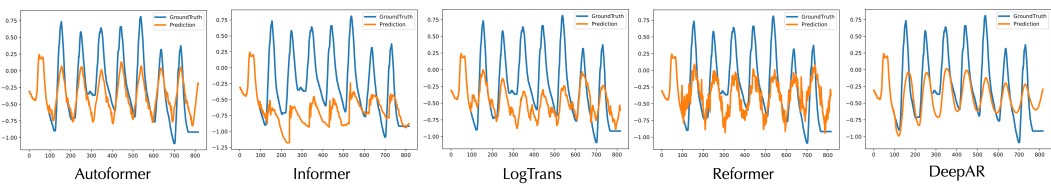

Figure 6: Prediction cases from the ETT dataset under the input-96-predict-720 **univariate** setting.

Table 6: Quantitative results with fluctuations under different prediction lengths $O$ for **multivariate** forecasting. We set the input length $I$ as 36 for ILI and 96 for the other datasets. A lower MSE or MAE indicates a better performance.

| Models | **Autoformer** | | Informer[14] | | LogTrans[9] | | Reformer[7] | |
|---|---|---|---|---|---|---|---|---|
| Metric | MSE | MAE | MSE | MAE | MSE | MAE | MSE | MAE |
| ETT 96 | **0.255**±0.020 | **0.339**±0.020 | 0.365±0.062 | 0.453±0.047 | 0.768±0.071 | 0.642±0.020 | 0.658±0.121 | 0.619±0.021 |
| ETT 192 | **0.281**±0.027 | **0.340**±0.025 | 0.533±0.109 | 0.563±0.050 | 0.989±0.124 | 0.757±0.049 | 1.078±0.106 | 0.827±0.012 |
| ETT 336 | **0.339**±0.018 | **0.372**±0.015 | 1.363±0.173 | 0.887±0.056 | 1.334±0.168 | 0.872±0.054 | 1.549±0.146 | 0.972±0.015 |
| ETT 720 | **0.422**±0.015 | **0.419**±0.010 | 3.379±0.143 | 1.388±0.037 | 3.048±0.140 | 1.328±0.023 | 2.631±0.126 | 1.242±0.014 |
| Electricity 96 | **0.201**±0.003 | **0.317**±0.004 | 0.274±0.004 | 0.368±0.003 | 0.258±0.002 | 0.357±0.002 | 0.312±0.003 | 0.402±0.004 |
| Electricity 192 | **0.222**±0.003 | **0.334**±0.004 | 0.296±0.009 | 0.386±0.007 | 0.266±0.005 | 0.368±0.004 | 0.348±0.004 | 0.433±0.005 |
| Electricity 336 | **0.231**±0.006 | **0.338**±0.004 | 0.300±0.007 | 0.394±0.004 | 0.280±0.006 | 0.380±0.001 | 0.350±0.004 | 0.433±0.003 |
| Electricity 720 | **0.254**±0.007 | **0.361**±0.008 | 0.373±0.034 | 0.439±0.024 | 0.283±0.003 | 0.376±0.002 | 0.340±0.002 | 0.420±0.002 |
| Exchange 96 | **0.197**±0.019 | **0.323**±0.012 | 0.847±0.150 | 0.752±0.060 | 0.968±0.177 | 0.812±0.027 | 1.065±0.070 | 0.829±0.013 |
| Exchange 192 | **0.300**±0.020 | **0.369**±0.016 | 1.204±0.149 | 0.895±0.061 | 1.040±0.232 | 0.851±0.029 | 1.188±0.041 | 0.906±0.008 |
| Exchange 336 | **0.509**±0.041 | **0.524**±0.016 | 1.672±0.036 | 1.036±0.014 | 1.659±0.122 | 1.081±0.015 | 1.357±0.027 | 0.976±0.010 |
| Exchange 720 | **1.447**±0.084 | **0.941**±0.028 | 2.478±0.198 | 1.310±0.070 | 1.941±0.327 | 1.127±0.030 | 1.510±0.071 | 1.016±0.008 |
| Traffic 96 | **0.613**±0.028 | **0.388**±0.012 | 0.719±0.015 | 0.391±0.004 | 0.684±0.041 | 0.384±0.008 | 0.732±0.027 | 0.423±0.025 |
| Traffic 192 | **0.616**±0.042 | **0.382**±0.020 | 0.696±0.050 | 0.379±0.023 | 0.685±0.055 | 0.390±0.021 | 0.733±0.013 | 0.420±0.011 |
| Traffic 336 | **0.622**±0.016 | **0.337**±0.011 | 0.777±0.009 | 0.420±0.003 | 0.733±0.069 | 0.408±0.026 | 0.742±0.012 | 0.420±0.008 |
| Traffic 720 | **0.660**±0.025 | **0.408**±0.015 | 0.864±0.026 | 0.472±0.015 | 0.717±0.030 | 0.396±0.010 | 0.755±0.023 | 0.423±0.014 |
| Weather 96 | **0.266**±0.007 | **0.336**±0.006 | 0.300±0.013 | 0.384±0.013 | 0.458±0.143 | 0.490±0.038 | 0.689±0.042 | 0.596±0.019 |
| Weather 192 | **0.307**±0.024 | **0.367**±0.022 | 0.598±0.045 | 0.544±0.028 | 0.658±0.151 | 0.589±0.032 | 0.752±0.048 | 0.638±0.029 |
| Weather 336 | **0.359**±0.035 | **0.395**±0.031 | 0.578±0.024 | 0.523±0.016 | 0.797±0.034 | 0.652±0.019 | 0.639±0.030 | 0.596±0.021 |
| Weather 720 | **0.419**±0.017 | **0.428**±0.014 | 1.059±0.096 | 0.741±0.042 | 0.869±0.045 | 0.675±0.093 | 1.130±0.084 | 0.792±0.055 |
| ILI 24 | **3.483**±0.107 | **1.287**±0.018 | 5.764±0.354 | 1.677±0.080 | 4.480±0.313 | 1.444±0.033 | 4.400±0.117 | 1.382±0.021 |
| ILI 36 | **3.103**±0.139 | **1.148**±0.025 | 4.755±0.248 | 1.467±0.067 | 4.799±0.251 | 1.467±0.023 | 4.783±0.138 | 1.448±0.023 |
| ILI 48 | **2.669**±0.151 | **1.085**±0.037 | 4.763±0.295 | 1.469±0.059 | 4.800±0.233 | 1.468±0.021 | 4.832±0.122 | 1.465±0.016 |
| ILI 60 | **2.770**±0.085 | **1.125**±0.019 | 5.264±0.237 | 1.564±0.044 | 5.278±0.231 | 1.560±0.014 | 4.882±0.123 | 1.483±0.016 |

## 5.4 Main Results with Standard Deviations

To get more robust experimental results, we repeat each experiment three times. The results are shown without standard deviations in the main text due to the limited pages. Table 6 shows the main results with standard deviations.

## 6 COVID-19: Case Study

We also apply our model to the COVID-19 real-world data [4]. This dataset contains the data collected from countries, including the number of confirmed deaths and recovered patients of COVID-19 recorded daily from January 22, 2020, to May 20, 2021. We select two anonymous countries in Europe for the experiments. The data is split into training, validation and test set in chronological order following the ratio of 7:1:2 and normalized. Note that this problem is quite challenging because the training data is limited.

### 6.1 Quantitative Results

We still follow the long-term forecasting task and let the model predict the next week, half month, full month respectively. The prediction lengths are 1, 2.1, 4.3 times the input length. As shown in Table 7, Autoformer still keeps the state-of-the-art accuracy under the **limited data** and **short input** situation.

### 6.2 Showcases

As shown in Figure 7, compared to other models, our Autoformer can accurately predict the peaks and troughs at the beginning and can almost predict the exact value in the long-term future. The forecasting of extreme values and long-term trends are essential to epidemic prevention and control.

Table 7: Quantitative results for COVID-19 data. We set the input length $I$ as 7, which means that the data in one week. The prediction length $O$ is in $\{7, 15, 30\}$, which represents a week, half a month, a month respectively. A lower MSE or MAE indicates a better prediction.

| Models | | **Autoformer** | | Informer[14] | | LogTrans[9] | | Reformer[7] | | Transformer[12] | |
|---|---|---|---|---|---|---|---|---|---|---|---|
| Metric | | MSE | MAE | MSE | MAE | MSE | MAE | MSE | MAE | MSE | MAE |
| Country 1 | 7 | **0.110** | **0.213** | 0.168 | 0.323 | 0.190 | 0.311 | 0.219 | 0.312 | 0.156 | 0.254 |
| | 15 | **0.168** | **0.264** | 0.443 | 0.482 | 0.229 | 0.361 | 0.276 | 0.403 | 0.289 | 0.382 |
| | 30 | **0.261** | **0.319** | 0.443 | 0.482 | 0.311 | 0.356 | 0.276 | 0.403 | 0.362 | 0.444 |
| Country 2 | 7 | **1.747** | **0.891** | 1.806 | 0.969 | 1.834 | 1.013 | 2.403 | 1.071 | 1.798 | 0.955 |
| | 15 | **1.749** | **0.905** | 1.842 | 0.969 | 1.829 | 1.004 | 2.627 | 1.111 | 1.830 | 0.999 |
| | 30 | **1.749** | **0.903** | 2.087 | 1.116 | 2.147 | 1.106 | 3.316 | 1.267 | 2.190 | 1.172 |

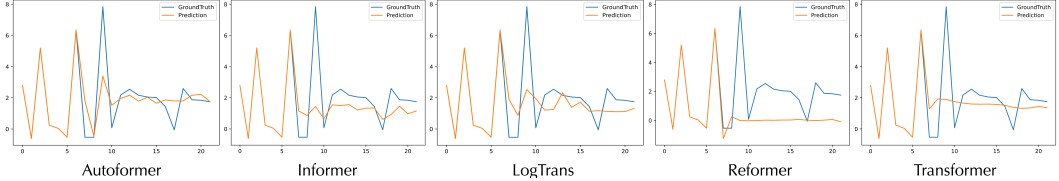

Figure 7: Prediction cases from the second country of COVID-19 under the input-7-predict-15 setting.

# 7 Autoformer: Implementation Details

## 7.1 Model Design

We provide the pseudo-code of Autoformer and Auto-Correlation mechanism in Algorithms 1 and 2 respectively. The tensor shapes and hyper-parameter settings are also included. Besides the above standard version, we speed up the Auto-Correlation to a batch-normalization-style block for efficiency, namely *speedup version*. **All the experiment results of this paper are from the speedup version.** Here are the implementation details.

**Speedup version**   Note that the `gather` operation in Algorithm 2 is not memory-access friendly. We borrow the design of *batch normalization* [6] to speedup the Auto-Correlation mechanism. We separate the whole procedure as the training phase and the inference phase. Because of the property of the linear layer, the channels of deep representations are equivalent. Thus, we reduce the channel and head dimension for both the training and inference phases. Especially for the training phase, we average the autocorrelation within a batch to simplify the learned lags. This design speeds up Auto-Correlation and performs as normalization to obtain a global judgment of the learned lags because the series within a batch are samples from the same time-series dataset. The pseudo-code for the training phase is presented in Algorithm 3. For the testing phase, we still use the `gather` operation with respect to the simplified lags, which is more memory-access friendly than the standard version. The pseudo-code for the inference phase is presented in Algorithm 4.

**Complexity analysis**   Our model provides the series-wise aggregation for $\lfloor c \times \log L \rfloor$ delayed length-$L$ series. Thus, the complexity is $\mathcal{O}(L \log L)$ for both the standard version and the speedup version. However, the latter is faster because it is more memory-access friendly.

## 7.2 Experiment Details

All these transformer-based models are built with two encoder layers and one decoder layer for the sake of the fair comparison in performance and efficiency, including Informer [14], Reformer [7], LogTrans [9] and canonical Transformer [12]. Besides, all these models adopt the embedding method and the one-step generation strategy as Informer [14]. Note that our proposed series-wise aggregation can provide enough sequential information. Thus, we do not employ the position embedding as other baselines but keep the value embedding and time stamp embedding.

**Algorithm 1** Overall Autoformer Procedure

**Input:** Input past time series $\mathcal{X}$; Input Length $I$; Predict length $O$; Data dimension $d$; Hidden state channel $d_{\text{model}}$; Encoder layers number $N$; Decoder layers number $M$; Moving average window size $k$. Technically, we set $d_{\text{model}}$ as 512, $N$ as 2, $M$ as 1, $k$ as 25.

1: $\mathcal{X}_{\text{ens}}, \mathcal{X}_{\text{ent}} = \texttt{SeriesDecomp}(\mathcal{X}_{\frac{I}{2}:I})$          $\triangleright\ \mathcal{X} \in \mathbb{R}^{I \times d}, \mathcal{X}_{\text{ens}}, \mathcal{X}_{\text{ent}} \in \mathbb{R}^{\frac{I}{2} \times d}$

2: $\mathcal{X}_0, \mathcal{X}_{\text{mean}} = \texttt{Zeros}([O, d]), \texttt{Repeat}\Big(\texttt{Mean}(\mathcal{X}_{\frac{I}{2}:I}, \texttt{dim=0}), \texttt{dim=0}\Big)$     $\triangleright\ \mathcal{X}_0, \mathcal{X}_{\text{mean}} \in \mathbb{R}^{O \times d}$

3: $\mathcal{X}_{\text{des}}, \mathcal{X}_{\text{det}} = \texttt{Concat}(\mathcal{X}_{\text{ens}}, \mathcal{X}_0), \texttt{Concat}(\mathcal{X}_{\text{ent}}, \mathcal{X}_{\text{mean}})$     $\triangleright\ \mathcal{X}_{\text{des}}, \mathcal{X}_{\text{det}} \in \mathbb{R}^{(\frac{I}{2}+O) \times d}$

4: $\mathcal{X}_{\text{en}}^0 = \texttt{Embed}(\mathcal{X})$          $\triangleright\ \mathcal{X}_{\text{en}}^0 \in \mathbb{R}^{I \times d_{\text{model}}}$

5: **for** $l$ **in** $\{1, \cdots, N\}$**:**          $\triangleright$ Autoformer Encoder

6:      $\mathcal{S}_{\text{en}}^{l,1}, \_ = \texttt{SeriesDecomp}\Big(\texttt{Auto-Correlation}(\mathcal{X}_{\text{en}}^{l-1}) + \mathcal{X}_{\text{en}}^{l-1}\Big)$    $\triangleright\ \mathcal{S}_{\text{en}}^{l,1} \in \mathbb{R}^{I \times d_{\text{model}}}$

7:      $\mathcal{S}_{\text{en}}^{l,2}, \_ = \texttt{SeriesDecomp}\Big(\texttt{FeedForward}(\mathcal{S}_{\text{en}}^{l,1}) + \mathcal{S}_{\text{en}}^{l,1}\Big)$    $\triangleright\ \mathcal{S}_{\text{en}}^{l,2} \in \mathbb{R}^{I \times d_{\text{model}}}$

8:      $\mathcal{X}_{\text{en}}^l = \mathcal{S}_{\text{en}}^{l,2}$          $\triangleright\ \mathcal{X}_{\text{en}}^l \in \mathbb{R}^{I \times d_{\text{model}}}$

9: **End for**

10: $\mathcal{X}_{\text{de}}^0 = \texttt{Embed}(\mathcal{X}_{\text{des}}), \mathcal{T}_{\text{de}}^0 = \mathcal{X}_{\text{det}},$       $\triangleright\ \mathcal{X}_{\text{de}}^0 \in \mathbb{R}^{(\frac{I}{2}+O) \times d_{\text{model}}}, \mathcal{T}_{\text{de}}^0 \in \mathbb{R}^{(\frac{I}{2}+O) \times d}$

11: **for** $l$ **in** $\{1, \cdots, M\}$**:**          $\triangleright$ Autoformer Decoder

12:      $\mathcal{S}_{\text{de}}^{l,1}, \mathcal{T}_{\text{de}}^{l,1} = \texttt{SeriesDecomp}\Big(\texttt{Auto-Correlation}(\mathcal{X}_{\text{de}}^{l-1}) + \mathcal{X}_{\text{de}}^{l-1}\Big)$

13:      $\mathcal{S}_{\text{de}}^{l,2}, \mathcal{T}_{\text{de}}^{l,2} = \texttt{SeriesDecomp}\Big(\texttt{Auto-Correlation}(\mathcal{S}_{\text{de}}^{l,1}, \mathcal{X}_{\text{en}}^{N}) + \mathcal{S}_{\text{de}}^{l,1}\Big)$

14:      $\mathcal{S}_{\text{de}}^{l,3}, \mathcal{T}_{\text{de}}^{l,3} = \texttt{SeriesDecomp}\Big(\texttt{FeedForward}(\mathcal{S}_{\text{de}}^{l,2}) + \mathcal{S}_{\text{de}}^{l,2}\Big)$    $\triangleright\ \mathcal{S}_{\text{de}}^{l,\cdot}, \mathcal{T}_{\text{de}}^{l,\cdot} \in \mathbb{R}^{(\frac{I}{2}+O) \times d_{\text{model}}}$

15:      $\mathcal{T}_{\text{de}}^l = \mathcal{T}_{\text{de}}^{l-1} + \texttt{MLP}(\mathcal{T}_{\text{de}}^{l,1}) + \texttt{MLP}(\mathcal{T}_{\text{de}}^{l,2}) + \texttt{MLP}(\mathcal{T}_{\text{de}}^{l,3})$    $\triangleright\ \mathcal{T}_{\text{de}}^l \in \mathbb{R}^{(\frac{I}{2}+O) \times d}$

16:      $\mathcal{X}_{\text{de}}^l = \mathcal{S}_{\text{de}}^{l,3}$          $\triangleright\ \mathcal{X}_{\text{de}}^l \in \mathbb{R}^{(\frac{I}{2}+O) \times d_{\text{model}}}$

17: **End for**

18: $\mathcal{X}_{\text{pred}} = \texttt{MLP}(\mathcal{X}_{\text{de}}^M) + \mathcal{T}_{\text{de}}^M$          $\triangleright\ \mathcal{X}_{\text{pred}} \in \mathbb{R}^{(\frac{I}{2}+O) \times d_{\text{model}}}$

19: **Return** $\mathcal{X}_{\text{pred}\ \frac{I}{2}:\frac{I}{2}+O}$          $\triangleright$ Return the prediction results

---

**Algorithm 2** Auto-Correlation (multi-head standard version for a batch of data)

**Input:** Queries $\mathcal{Q} \in \mathbb{R}^{B \times L \times d_{\text{model}}}$; Keys $\mathcal{K} \in \mathbb{R}^{B \times S \times d_{\text{model}}}$; Values $\mathcal{V} \in \mathbb{R}^{B \times S \times d_{\text{model}}}$; Number of heads $h$; Hidden state channel $d_{\text{model}}$; Hyper-parameter $c$. We set $d_{\text{model}}$ as 512, $h$ as 8, $1 \le c \le 3$.

1: $\mathcal{K}, \mathcal{V} = \texttt{Resize}(\mathcal{K}), \texttt{Resize}(\mathcal{V})$    $\triangleright$ Resize is truncation or zero filling. $\mathcal{K}, \mathcal{V} \in \mathbb{R}^{B \times L \times d_{\text{model}}}$

2: $\mathcal{Q}, \mathcal{K}, \mathcal{V} = \texttt{Reshape}(\mathcal{Q}), \texttt{Reshape}(\mathcal{K}), \texttt{Reshape}(\mathcal{V})$     $\triangleright\ \mathcal{Q}, \mathcal{K}, \mathcal{V} \in \mathcal{R}^{L \times h \times \frac{d_{\text{model}}}{h}}$

3: $\mathcal{Q} = \texttt{FFT}(\mathcal{Q}, \texttt{dim=0}), \mathcal{K} = \texttt{FFT}(\mathcal{K}, \texttt{dim=0}),$     $\triangleright\ \mathcal{Q}, \mathcal{K} \in \mathbb{C}^{B \times L \times h \times \frac{d_{\text{model}}}{h}}$

4: $\texttt{Corr} = \texttt{IFFT}\Big(\mathcal{Q} \times \texttt{Conj}(\mathcal{K}), \texttt{dim=0}\Big)$     $\triangleright$ Autocorrelation $\texttt{Corr} \in \mathbb{R}^{B \times L \times h \times \frac{d_{\text{model}}}{h}}$

5: $\texttt{W}_{\text{topk}}, \texttt{I}_{\text{topk}} = \texttt{Topk}(\texttt{Corr}, \lfloor c \times \log L \rfloor, \texttt{dim=0})$    $\triangleright$ Largest weights $\texttt{W}_{\text{topk}}$ and their indices $\texttt{I}_{\text{topk}}$

6: $\texttt{W}_{\text{topk}} = \texttt{Softmax}(\texttt{W}_{\text{topk}}, \texttt{dim=0})$     $\triangleright\ \texttt{W}_{\text{topk}}, \texttt{I}_{\text{topk}} \in \mathbb{R}^{B \times (\lfloor c \times \log L \rfloor) \times h \times \frac{d_{\text{model}}}{h}}$

7: $\texttt{Index} = \texttt{Repeat}\Big(\texttt{arange}(L)\Big)$     $\triangleright$ Initialize series indices. $\texttt{Index} \in \mathbb{R}^{B \times L \times h \times \frac{d_{\text{model}}}{h}}$

8: $\mathcal{V} = \texttt{Repeat}(\mathcal{V})$     $\triangleright\ \mathcal{V} \in \mathbb{R}^{B \times (2L) \times h \times \frac{d_{\text{model}}}{h}}$

9: $\mathcal{R} = \Big[\texttt{W}_{\text{topk}\,i,:,:} \times \texttt{gather}\Big(\mathcal{V}, (\texttt{I}_{\text{topk}\,i,:,:} + \texttt{Index})\Big)\ \textbf{for}\ i\ \textbf{in}\ \texttt{range}(\lfloor c \times \log L \rfloor)\Big]$   $\triangleright$ Aggregation

10: $\mathcal{R} = \texttt{Sum}\Big(\texttt{Stack}(\mathcal{R}, \texttt{dim=0}), \texttt{dim=0}\Big)$     $\triangleright\ \mathcal{R} \in \mathbb{R}^{B \times L \times h \times \frac{d_{\text{model}}}{h}}$

11: **Return** $\mathcal{R}$          $\triangleright$ Return transformed results

**Algorithm 3** Auto-Correlation (multi-head **speedup version** for the **training phase**)

---

**Input:** Queries $\mathcal{Q} \in \mathbb{R}^{B \times L \times d_{\text{model}}}$; Keys $\mathcal{K} \in \mathbb{R}^{B \times S \times d_{\text{model}}}$; Values $\mathcal{V} \in \mathbb{R}^{B \times S \times d_{\text{model}}}$; Number of heads $h$; Hidden state channel $d_{\text{model}}$; Hyper-parameter $c$. We set $d_{\text{model}}$ as 512, $h$ as 8, $1 \leq c \leq 3$.

1: $\mathcal{K}, \mathcal{V} = \texttt{Resize}(\mathcal{K}), \texttt{Resize}(\mathcal{V})$      $\triangleright$ Resize is truncation or zero filling. $\mathcal{K}, \mathcal{V} \in \mathbb{R}^{B \times L \times d_{\text{model}}}$

2: $\mathcal{Q}, \mathcal{K}, \mathcal{V} = \texttt{Reshape}(\mathcal{Q}), \texttt{Reshape}(\mathcal{K}), \texttt{Reshape}(\mathcal{V})$      $\triangleright$ $\mathcal{Q}, \mathcal{K}, \mathcal{V} \in \mathcal{R}^{B \times L \times h \times \frac{d_{\text{model}}}{h}}$

3: $\mathcal{Q} = \texttt{FFT}(\mathcal{Q}, \texttt{dim=0}), \mathcal{K} = \texttt{FFT}(\mathcal{K}, \texttt{dim=0}),$      $\triangleright$ $\mathcal{Q}, \mathcal{K} \in \mathbb{C}^{B \times L \times h \times \frac{d_{\text{model}}}{h}}$

4: $\texttt{Corr} = \texttt{IFFT}\Big(\mathcal{Q} \times \texttt{Conj}(\mathcal{K}), \texttt{dim=0}\Big)$      $\triangleright$ Autocorrelation $\texttt{Corr} \in \mathbb{R}^{B \times L \times h \times \frac{d_{\text{model}}}{h}}$

5: $\texttt{Corr} = \texttt{Mean}(\texttt{Corr}, dim = 0, 2, 3)$      $\triangleright$ Simplify lags. $\texttt{Corr} \in \mathbb{R}^{L}$

6: $\text{W}_{\text{topk}}, \text{I}_{\text{topk}} = \texttt{Topk}(\texttt{Corr}, \lfloor c \times \log L \rfloor, \texttt{dim=0})$      $\triangleright$ Largest weights $\text{W}_{\text{topk}}$ and their indices $\text{I}_{\text{topk}}$

7: $\text{W}_{\text{topk}} = \texttt{Softmax}(\text{W}_{\text{topk}}, \texttt{dim=0})$      $\triangleright$ $\text{W}_{\text{topk}}, \text{I}_{\text{topk}} \in \mathbb{R}^{(\lfloor c \times \log L \rfloor)}$

8: $\mathcal{R} = \Big[\text{W}_{\text{topk}_{i,:,:}} \times \texttt{Roll}(\mathcal{V}, \text{I}_{\text{topk}\,i,:,:}, \texttt{dim=1}) \textbf{ for } i \textbf{ in } \texttt{range}(\lfloor c \times \log L \rfloor)\Big]$      $\triangleright$ Aggregation

9: $\mathcal{R} = \texttt{Sum}\Big(\texttt{Stack}(\mathcal{R}, \texttt{dim=0}), \texttt{dim=0}\Big)$      $\triangleright$ $\mathcal{R} \in \mathbb{R}^{L \times h \times \frac{d_{\text{model}}}{h}}$

10: **Return** $\mathcal{R}$      $\triangleright$ Return transformed results

---

**Algorithm 4** Auto-Correlation (multi-head **speedup version** for the **inference phase**)

---

**Input:** Queries $\mathcal{Q} \in \mathbb{R}^{B \times L \times d_{\text{model}}}$; Keys $\mathcal{K} \in \mathbb{R}^{B \times S \times d_{\text{model}}}$; Values $\mathcal{V} \in \mathbb{R}^{B \times S \times d_{\text{model}}}$; Number of heads $h$; Hidden state channel $d_{\text{model}}$; Hyper-parameter $c$. We set $d_{\text{model}}$ as 512, $h$ as 8, $1 \leq c \leq 3$.

1: $\mathcal{K}, \mathcal{V} = \texttt{Resize}(\mathcal{K}), \texttt{Resize}(\mathcal{V})$      $\triangleright$ Resize is truncation or zero filling. $\mathcal{K}, \mathcal{V} \in \mathbb{R}^{B \times L \times d_{\text{model}}}$

2: $\mathcal{Q}, \mathcal{K}, \mathcal{V} = \texttt{Reshape}(\mathcal{Q}), \texttt{Reshape}(\mathcal{K}), \texttt{Reshape}(\mathcal{V})$      $\triangleright$ $\mathcal{Q}, \mathcal{K}, \mathcal{V} \in \mathcal{R}^{L \times h \times \frac{d_{\text{model}}}{h}}$

3: $\mathcal{Q} = \texttt{FFT}(\mathcal{Q}, \texttt{dim=0}), \mathcal{K} = \texttt{FFT}(\mathcal{K}, \texttt{dim=0}),$      $\triangleright$ $\mathcal{Q}, \mathcal{K} \in \mathbb{C}^{B \times L \times h \times \frac{d_{\text{model}}}{h}}$

4: $\texttt{Corr} = \texttt{IFFT}\Big(\mathcal{Q} \times \texttt{Conj}(\mathcal{K}), \texttt{dim=0}\Big)$      $\triangleright$ Autocorrelation $\texttt{Corr} \in \mathbb{R}^{B \times L \times h \times \frac{d_{\text{model}}}{h}}$

5: $\texttt{Corr} = \texttt{Mean}(\texttt{Corr}, dim = 0, 2, 3)$      $\triangleright$ Simplify lags. $\texttt{Corr} \in \mathbb{R}^{L}$

6: $\text{W}_{\text{topk}}, \text{I}_{\text{topk}} = \texttt{Topk}(\texttt{Corr}, \lfloor c \times \log L \rfloor, \texttt{dim=0})$      $\triangleright$ Largest weights $\text{W}_{\text{topk}}$ and their indices $\text{I}_{\text{topk}}$

7: $\text{W}_{\text{topk}} = \texttt{Softmax}(\text{W}_{\text{topk}}, \texttt{dim=0})$      $\triangleright$ $\text{W}_{\text{topk}}, \text{I}_{\text{topk}} \in \mathbb{R}^{(\lfloor c \times \log L \rfloor)}$

8: $\texttt{Index} = \texttt{Repeat}\Big(\texttt{arange}(L)\Big)$      $\triangleright$ Initialize series indices. $\texttt{Index} \in \mathbb{R}^{B \times L \times h \times \frac{d_{\text{model}}}{h}}$

9: $\mathcal{V} = \texttt{Repeat}(\mathcal{V})$      $\triangleright$ $\mathcal{V} \in \mathbb{R}^{B \times (2L) \times h \times \frac{d_{\text{model}}}{h}}$

10: $\mathcal{R} = \Big[\text{W}_{\text{topk}_{i,:,:}} \times \texttt{gather}\Big(\mathcal{V}, (\text{I}_{\text{topk}\,i,:,:} + \texttt{Index})\Big) \textbf{ for } i \textbf{ in } \texttt{range}(\lfloor c \times \log L \rfloor)\Big]$    $\triangleright$ Aggregation

11: $\mathcal{R} = \texttt{Sum}\Big(\texttt{Stack}(\mathcal{R}, \texttt{dim=0}), \texttt{dim=0}\Big)$      $\triangleright$ $\mathcal{R} \in \mathbb{R}^{B \times L \times h \times \frac{d_{\text{model}}}{h}}$

12: **Return** $\mathcal{R}$      $\triangleright$ Return transformed results

---

## 8 Broader Impact

**Real-world applications**   Our proposed Autoformer focuses on the long-term time series forecasting problem, which is a valuable and urgent demand in extensive applications. Our method achieves consistent state-of-the-art performance in five real-world applications: energy, traffic, economics, weather and disease. In addition, we provide the case study of the COVID-19 dataset. Thus, people who work in these areas may benefit greatly from our work. We believe that better time series forecasting can help our society make better decisions and prevent risks in advance for various fields.

**Academic research**   In this paper, we take the ideas from classic time series analysis and stochastic process theory. We innovate a general deep decomposition architecture with a novel Auto-Correlation mechanism, which is a worthwhile addition to time series forecasting models. Code is available at this repository: `https://github.com/thuml/Autoformer`.

**Model Robustness**   Based on the extensive experiments, we do not find exceptional failure cases. Autoformer even provides good performance and long-term robustness in the *Exchange* dataset that does not present obvious periodicity. Autoformer can progressively get purer series components by the inner decomposition block and make it easy to discover the deeply hidden periodicity. However, if the data is random or with extremely weak temporal coherence, Autoformer and any other models may degenerate because the series is with poor predictability [3].

Our work only focuses on the scientific problem, so there is no potential ethical risk.