# OpenReview forum: "Autoformer: Decomposition Transformers with Auto-Correlation for Long-Term Series Forecasting"
_NeurIPS.cc/2021/Conference — NeurIPS 2021 Poster_

### Official Review · Reviewer_zbiv · 2021-07-05

**Rating:** 6
**Confidence:** 3

**Summary:**

This paper introduces a new approach for long-term time-series forecasting that is based on decomposition and auto-correlation. Essentially, the authors define a series decomoposition block which extracts the mean (trend-cyclical) and subtracts it from the series, obtaining the seasonal part. In addition, the authors design an auto-correlation component that identifies local structures across periods. The method is evaluated on six benchmark datasets and in comparison to several baseline approaches. The presented results show improvements over existing work.

**Limitations And Societal Impact:**

See above

**Main Review:**

The authors of this submission propose to modify the transformer module to account for its limiting time/space computational complexity. To this end, they propose two components---decomposition and autocorrelation, which work together to yield improved long-term forecasts. To the best of my knowledge, this idea is novel. The exposition of the paper is mostly clear, however, Subsec. 3.2 should be explained better as it is not entirely clear how it operates and implemented. Similarly, the descriptions in Subsec 3.1 could be improved, but they are somewhat better than 3.2. There are some undefined operations such as AvgPool, and it was not clear whether AutoCorrelation takes one argument or two.

Overall, the proposed method is interesting, and the reported results are impressive. However, I would have liked to see stronger ties to the theory and practice of decompositions and auto-correlations in this work. Unfortunately, the current version facilitates these components purely on the usage level, and it is not entirely clear why the results are better over other strong baselines. For instance, what happens if there are minimal to no local recurrences of self-structures? Would the autocorrelation help at all? What happens when ones moves to a higher-dimensional setting where signals can be rotated/stretched---would the autocorrelation component be able to account for that?

The limitations of the technique were not clearly described. For instance, does the model depend on the length of the series (L) as it presumably takes average across this dimension? How one decided if to use additive or multiplicative decompositions?


**Time Spent Reviewing:**

4

---

> ### Author Response · Authors · 2021-08-08
> **Response to Reviewer zbiv**
>
> We would like to sincerely thank Reviewer zbiv for providing the detailed review and insightful suggestions. We will further clarify the points in the revision.
>
> **Q1:** Details about our method.
>
> (1) Section 3.1: We have provided the implementation details in $\underline{\text{Section 8 of supplementary material}}$. This section describes the method from a tensor view.
>
> (2) Section 3.2: We show the tensor shape of each operation with the gray equations in $\underline{\text{Figure 2 (left) of main paper}}$. The first dimension represents the temporal size, and the second is for the channel size. For example, $L\times C$ denotes the tensor length is $L$ and channel number is $C$. The "resize" operation is in $\underline{\text{line 93-96 of supplemental material}}$.
>
> (3) AvgPool: This operation is the $\texttt{nn.avgpool1d}$ in the deep learning framework PyTorch [23], which takes the average of tensor along the temporal dimension. We use the padding operation to keep the tensor length unchanged.
>
> (4) Auto-Correlation has the same usage as self-attention in Transformer [34]. Thus, it can be written as the function $\mathrm{AutoCorrelation}(Q,K,V)$ with three arguments: queries $Q$, keys $K$ and values $V$. For the encoder Auto-Correlation and the first Auto-Correlation in the decoder block, the three arguments are from the previous block's output. They can be formalized as $\mathrm{AutoCorrelation}(X_{\mathrm{en}}^{l-1},X_{\mathrm{en}}^{l-1},X_{\mathrm{en}}^{l-1})$ and $\mathrm{AutoCorrelation}(X_{\mathrm{de}}^{l-1},X_{\mathrm{de}}^{l-1},X_{\mathrm{de}}^{l-1})$ for the $l$th layer. As stated in $\underline{\text{line 178-179 of main paper}}$, for the second Auto-Correlation in the decoder block, the queries are from the previous block's output, while keys and values are from the encoder output. So it is $\mathrm{AutoCorrelation}(S_{\mathrm{de}}^{l,1},X_{\mathrm{en}}^{N},X_{\mathrm{en}}^{N})$ where $N$ is the number of encoder layers.
>
> Thanks for your suggestion. We will clarify these in the revision.
>
>
>
> **Q2:** The explanation of why the results are better over other strong baselines.
>
> Our paper focuses on the design of a basic deep model for long series forecasting. It is enlightened by classic methods and concepts and provides a non-trivial design. Note that it is not a theorem paper and it is hard to give a complete theorem for deep models. This is in parallel with the widely-used deep models such as ResNet in computer vision or Transformers [34] in natural language processing. Thus, we have explained the effect of decomposition architecture and Auto-Correlation from the empirical aspects.
>
> First, for the decomposition architecture, we have visualized the decomposition results of time series in $\underline{\text{Figure 4 of main paper}}$. This verifies that deeper decomposition can get more predictable components.
>
> Second, Auto-Correlation performs better than the self-attention family in the following two aspects:
>
> (1) **Discover the relevant information more sufficiently and precisely.** As shown in $\underline{\text{Figure 5 of main paper}}$, we visualize the learned dependencies from Auto-Correlation and various attentions. Auto-Correlation can capture the right sub-processes without omissions or errors. This means that without full connections, Auto-Correlation can utilize the series information even better.
>
> (2) **Aggregate the relevant information underlying the context of time series.** As shown in the intuitive comparison in $\underline{\text{Figure 3(d) of main paper}}$, Auto-Correlation aggregates the relevant information from sub-series  to sub-series. While, attentions aggregate the information point by point, regardless of the time series context.
>
>
>
> **Q3:** Discussion under various conditions mentioned by the reviewer.
>
> (1) There are minimal to no local recurrences of self-structure.
>
> We have provided some analysis for the Exchange dataset in $\underline{\text{Section 4.2 of supplementary material}}$. Exchange dataset does not exhibit strong periodicity. Autoformer can still produce a fairly good prediction. Note that Autoformer contains the decomposition architecture and the Auto-Correlation. These two parts will work seamlessly to deal with complex series, such as the mentioned situation.
>
> (2) The higher-dimensional setting where the signal can be rotated/stretched.
>
> Recently, Transformers [34] have shown greater power in learning long-range dependencies, such as GPT-3 [7] or ViT [11]. The learned attention maps of deep features are usually meaningful (e.g. ViT [11]).
>
> In Autoformer, $\underline{\text{Figure 5 of main paper}}$ shows **the learned lags of Auto-Correlation**. This visualization is for a raw series in the ETT dataset. Although the series is projected to a deep representation with 512 channels, Auto-Correlation can still find the periods well. This verifies the effectiveness under the higher-dimensional setting.
>
> In addition, we visualize the deep representation with 512 channels. The deep representation exhibits a similar periodicity to the raw series. Limited by the edit box of the OpenReview platform, we cannot paste the figure here, but we will add this in the revision.
>
>
>
> **Q4:** The limitations.
>
> We mostly employ the fixed input length for a clear comparison among different forecasting horizons. The input length $I$ is dataset-specific in real-world applications. And the best input length needs to be adjusted according to the data patterns. For example, on the Exchange dataset, the performance can be improved as the input length increases.
>
> | Autoformer (MSE \| MAE) | Input-96       | Input-192      | Input-336      | Input-720      |
> | ----------------------- | -------------- | -------------- | -------------- | -------------- |
> | Exchange (predict 336)  | 0.488 \| 0.510 | 0.420 \| 0.472 | 0.404 \| 0.460 | 0.392 \| 0.459 |
>
>
>
> **Q5:** Additive or multiplicative decompositions.
>
> Additive or multiplicative decompositions are two mainstream decomposition schemes. In this paper, we only consider the additive decompositions. This is because in deep learning models, addition is always used as a basic operator, like ResNet and Transformers. In contrast, the multiplication may result in the gradient explosion or vanishing problem and need a more subtle and sophisticated architecture design. We plan to further explore the multiplicative decompositions as our future work.

---

### Official Review · Reviewer_jDpZ · 2021-07-19

**Rating:** 7
**Confidence:** 4

**Summary:**

The paper proposes the two following novel architecture changes to improve transformer for long-term time-series forecasting
1) Time-series decomposition block, where trend and seasonality are progressively decomposed.
2) Auto-correlation attention mechanism, where attentions are computed in the frequency domain and aggregated by time delay aggregation, instead of computing attentions in the time domain and point-wise aggregated as in normal transformer architecture.

**Limitations And Societal Impact:**

I do not see the limitations or potential negative societal impact of the work.

**Main Review:**

The paper says that "*long-term forecasting* setting is to predict the long-term future given the short-term history".  To the best of my knowledge, this is not correct. "Long-term forecasting" depends solely on how far ahead we make the forecast, and should be independent of how much data we use to make the forecast. For example, in electricity forecasting, long-term forecasting usually means one month or up to a year ahead forecast. The paper uses the incorrect definition of *long-term forecasting* to justify its input data limit. All the experiments use only 96 data points as input, in order to claim that the proposed methods can make good forecasts where the length of the predicted series is severalfold of the length of the input series. I believe the setting makes most of the tasks unrealistic. For example, the Electricity dataset contains two-year hourly data (17520 data points), and the paper uses 4 days of data (96 data points) to forecast 2, 8, 14, 30 days ahead. This is a very unrealistic setting! Electricity consumption usually includes strong daily, weekly, and yearly patterns, with a long-term trend. Using 4 days of data input means that we are comparing models on their ability to capture the daily pattern only. A similar argument applies to other datasets (ETT, Traffic, Weather, Exchange) as well. It's even worse in the case of ETT dataset since it has 15 minutes resolution, so 96 data points represent a single day of data, while the whole dataset contains 2 years of data.

Looking at the series decomposition block, I believe that it can only effectively capture at most one seasonality because it only progressively produces one seasonal series. That could explain why the authors choose to limit the input data to 96 data points. Table 3 shows some evidence in favor of my hypothesis. **The MSE consistently increases when the input length increases**, regardless of whether auto-correlation or self-attention is used. This is very unexpected and shows that the model fails to work when the input data is long enough, maybe due to multiple seasonalities and trends.

It's also well-known that the correlogram of time-series with multiple seasonalities is very noisy due to interfering between seasonalities and patterns. So, I believe that the Auto-Correlation mechanism will not work for long time series with complex seasonalities, since picking top k lags that have the highest auto-correlation value would not include the periodicities we want. For example, most real-life time series will have recent lags (e.g., lag-1, lag-2) with the highest auto-correlation. It would be very helpful if the authors include the list of lags that the model pays attention to as a sanity check. To be clear, Figure 3.d. is just a demonstration of what the authors expect the auto-correlation mechanism to work, not how it actually works.

Another big concern of mine is that in the forecasting tasks, it's pretty simple to pick a window so that a particular model outperforms others. So it's highly recommended to do forecasting in a sliding window fashion so that the model gets tested at different time windows (windows could be overlapped or not, and the model could be retrained or not after each sliding). All experiments in the paper are conducted for a single testing window per task, which makes it really hard to conclude anything. Especially when the paper does not say anything about how they choose the testing window.


**Time Spent Reviewing:**

5

---

> ### Author Response · Authors · 2021-08-08
> **Response to Reviewer jDpZ**
>
> Many Thanks to Reviewer jDpZ for providing a detailed review and insightful questions.
>
> **Q1:** The definition of "Long-term forecasting".
>
> Thanks for your suggestion. We will provide more experiments about long inputs in $\underline{\text{Q3 of this rebuttal}}$, which will show that Autoformer also works in these cases when long-input is required to provide sufficient information. We will follow your suggestion and rephrase the definition as: "Long-term forecasting depends solely on how far ahead we make the forecast, regardless of the input length."
>
> Nonetheless, our paper considers a more challenging task: forecasting the long-term future with limited past data. This requires the model with stronger predictive ability. This is highly relevant in real-world applications of deep learning, where we have to trade-off between computational efficiency and information utilization. Here is the GPU memory cost for different input lengths.
>
> | Inputs Length (Predict 720) | Autoformer | Transformer |
> | -------------------- | ---------- | ----------- |
> | 96 (ETT)             | 4414 MB    | 5354 MB     |
> | 336 (ETT)            | 6000 MB    | 9912 MB     |
> | 720 (ETT)            | 8476 MB    | 16930 MB    |
>
> Because the forecasting horizon is usually fixed due to application's requirement, the long input is GPU prohibitive. Thus, the limited input is **quite meaningful for the real-world application of deep models**. As a common practice of deep learning for extremely long time series [16, 40], people perform **sampling** to obtain shorter series that reflect more complex periodicities.
>
>
>
> **Q2:** The experiment settings & model ability for different time-scale patterns.
>
> (1) Note that our model forecasts $O$ time points into the future based on the past $I$ time points. Thus, our model not only learns the patterns within the length-$I$ inputs, but also includes the patterns of the future length-$O$ outputs. Thus, our setting of input-96-predict-720 does **capture the various patterns and complex seasonalities in the sense of a 96+720 window**. This corresponds to the weekly pattern in the ETT dataset and the monthly pattern for the Traffic dataset.
>
> It is regular that deep models can learn the complex relation from a simple input to an intricate output, such as generative adversarial networks (GANs) that can generate a high-quality image based only on a simple noise as input. **Benefitting from the strong dependency reasoning ability of deep learning**, our models can predict the long-term future based on relatively short input.
>
> (2) To further address your concern, we change the sampling rate to evaluate the modeling ability for different time-scale patterns.
>
> - ILI dataset contains the data **collected weekly**. So the input-24-predict-60 results verify the long-term predictive ability of our model ($\underline{\text{Table 1 of main paper}}$).
> - Exchange dataset is **collected daily**. The input-96-predict-336 setting can also exhibit a **monthly or quarterly** modeling task ($\underline{\text{Table 1 of main paper}}$).
> - Following your opinion, we evaluate ETTh1 that is **hourly sampled** to verify the weekly patterns modeling. Autoformer performs well. We will include this result in the revision.
>
> | ETTh1 (MSE \| MAE) | Two weeks(336)->Two weeks(336) | Two weeks(336)->Four weeks(720) |
> | ------------------ | ------------------------------ | ------------------------------- |
> | Autoformer         | 0.537 \| 0.526                 | 0.637 \| 0.599                  |
> | Transformer        | 1.165 \| 0.882                 | 1.172 \| 0.915                  |
> | ARIMA              | 1.695｜0.910                   | 1.864 \| 0.971                  |
>
>
>
> **Q3:** The performance when the input data is long enough.
>
> To answer your question, we further investigate the situation and find it is **dataset-specific**. For the ETT dataset (with clear periodicity), an input length greater than the periodicity provides sufficient information (input length 96 to cover one day). Note that in $\underline{\text{Q2 of this rebuttal}}$ we have clarified that **Autoformer has the ability to capture complex seasonalities of long ranges in the sense of a 96+720 window**. In this case, longer inputs will force the model to handle more irrelevant information, which implicitly makes the forecasting task harder.
>
> While for the Exchange and ILI datasets (with complex seasonalities), it is often hard to determine how long the input length can provide sufficient information. In this case, enlarging the inputs will yield better results, as shown below.
>
> | Autoformer (MSE \| MAE) | Input-96     | Input-192    | Input-336    | Input-720    |
> | ----------------------- | ------------ | ------------ | ------------ | ------------ |
> | Exchange (predict 96)   | 0.134\|0.270 | 0.120\|0.259 | 0.114\|0.244 | 0.094\|0.225 |
> | Exchange (predict 192)  | 0.272\|0.374 | 0.210\|0.329 | 0.186\|0.314 | 0.184\|0.308 |
> | Exchange (predict 336)  | 0.488\|0.510 | 0.420\|0.472 | 0.404\|0.460 | 0.392\|0.459 |
> | Exchange (predict 720)  | 1.367\|0.901 | 1.212\|0.844 | 1.147\|0.838 | 1.142\|0.831 |
>
> | Autoformer (MSE \| MAE) | Input-24     | Input-36     | Input-48     | Input-60     |
> | ----------------------- | ------------ | ------------ | ------------ | ------------ |
> | ILI (predict 24)        | 3.825\|1.345 | 3.055\|1.189 | 2.738\|1.090 | 2.655\|1.040 |
> | ILI (predict 36)        | 3.319\|1.216 | 2.674\|1.024 | 2.544\|0.998 | 2.344\|0.995 |
> | ILI (predict 48)        | 2.854\|1.122 | 2.749\|1.095 | 2.717\|1.082 | 2.715\|1.075 |
> | ILI (predict 60)        | 3.227\|1.232 | 3.077\|1.219 | 2.822\|1.103 | 2.742\|1.014 |
>
> We follow your request and repeat the ablation experiment in $\underline{\text{Table 3 of main paper}}$ for the Exchange dataset. As shown below, when complex seasonalities are present, the performance of full attention is also improved as the input length increases.
>
> | Exchange (MSE \|MAE)           | Input-96       | Input-192      | Input-336      |
> | ------------------------------ | -------------- | -------------- | -------------- |
> | Auto-Correlation (predict 336) | 0.488 \| 0.510 | 0.420 \| 0.472 | 0.404 \| 0.460 |
> | Full Attention (predict 336)   | 0.632 \| 0.584 | 0.588 \| 0.572 | 0.546 \| 0.558 |
>
> All these results highlight that the behavior with respect to the input length is shared by well-establilished deep models (Transformers, Informer, Reformer, etc.), which **shall not be deemed a special limitation** of Autoformer.
>
>
>
> **Q4:** Auto-Correlation for complex seasonalities. Show the list of learned lags.
>
> (1) We only use $\underline{\text{Figure 3d of main paper}}$ for intuitive explanation. We also **visualized the learned lags** in $\underline{\text{Figure 5 of main paper}}$. This is for a real-world series from ETT dataset. Autoformer can capture all the similar processes among periods and does not only focus on the recent lags (e.g. lag-1 or lag-2).
>
> (2) To address your concern about complex seasonalities, we also visualize the learned lags in **daily collected** Exchange dataset that has complex seasonalities. Under the input-365-predict-365 setting, the learned top-10 lags are \{$312, 98, 97, 96, 99, 95, 311, 313, 126, 124$\}. These lags are close to \{$90, 120, 305(=365-60)$\} and show the monthly and quarterly seasonalities. We will add this in the revision, to complete Figure 5.
>
> (3) Furthermore, we construct an even complex setting. We add the ETT data (**collected every 15 minutes**) and the resampled ETT data (**collected every 1 hour**), regardless of the mismatch of time stamps. This mixed dataset contains multiple seasonalities with length 24 and 96. Autoformer can still perform well under this complex seasonalities. The top-10 learned lags are \{$287, 3, 97, 98, 291, 7, 10, 98, 95, 290$\}. These are close to \{$0, 96, 288$\} and match our expectation.
>
> | **Input-336-predict-336**     | **MSE** | **MAE** |
> | ----------------------------- | ------- | ------- |
> | Autoformer (Auto-Correlation) | 0.409   | 0.506   |
> | Transformer (Full Attention)  | 0.644   | 0.641   |
> | ARIMA                         | 0.857   | 0.785   |
>
>
>
> **Q5:** Sliding window fashion for evaluation.
>
> We have mentioned that we exactly follow the standard protocol in $\underline{\text{line 213 of main paper}}$. The standard protocol means **"rolling prediction without retraining"**. It uses a window of time points as the input and forecasts the future within the next window. The windows for training are randomly sampled from the train set. And we roll the window in the test set for evaluation.
>
> Specifically, we slide the window with length-$I$ in the test set as the model input and evaluate on next time window with length-$O$. Since the input window keeps sliding along the test set, we do test our model at different times. This is the common usage for many deep learning models, such as Informer [40], LogTrans [16]. In addition, we also evaluate on different forecasting horizons $O$ corresponding to different window sizes ($\underline{\text{Table 1 of main paper}}$). Thus, **it is a meaningful and sufficient comparison.**
>
> Corresponding to the classic method ARIMA, $p$ is the order (number of time lags) of the autoregressive part of ARIMA model. ARIMA will fit the parameters in the data with length-$I$ and then **slide** the autoregressive area with length $p$ to finish the next $O$ points prediction. In contrast, deep learning models finish this phase (from past $I$ to future $O$) in **one step**.
>
>
>
> **Q6:** The limitations.
>
> Our original paper did not explore the performance with the multi-scale inputs generated by different sampling rates. Also, the hyper-parameter $k = c \times \lfloor\log L\rfloor$ is dataset-specific and influenced by the data seasonalities. We will complete the hyperparameter analysis in $\underline{\text{Figure 1 of supplemental material}}$.
>
>
>
> Thanks for your valuable suggestions. We will incorporate all the responses in the future version.

---

> > ### Comment · Reviewer_jDpZ · 2021-08-10
> > **The response from authors resolves most of my concerns**
> >
> > Thanks a lot for clarifying the experiment setup (Q2, Q5) and providing extra data points to show the ability of AutoFormer in handling longer input window (Q3) and capturing multi-seasonalities (Q4). I have changed my Rating to "7: Good paper, accept" accordingly.
> > It would also be great if the authors can add to the paper simple clarifications about "rolling prediction without retraining", learned lags, memory constraints.

---

> > > ### Author Response · Authors · 2021-08-10
> > > **Response to Reviewer jDpZ**
> > >
> > > We'd like to thank Reviewer jDpZ again for providing an impressively insightful pre-rebuttal review, which has enabled us to make an effective response. We'd also thank you for carefully judging our feedback and acknowledging our work in the final review. We will guarantee to further clarify the points that you mentioned in the revision, including the evaluation protocol, learned lags, memory constraints.

---

### Official Review · Reviewer_GKTE · 2021-07-22

**Rating:** 7
**Confidence:** 4

**Summary:**

This paper focuses on the long-term forecasting problem of time series. This work introduces the traditional idea of decomposing time series into seasonality and trend-cycle components into deep Transformer architecture. This work also proposes an auto-correlation mechanism that replaces the original self-attention in Transformers.  Further, they improve the computation efficiency by computing power spectral density. Experiments on several datasets show that the proposed method achieves better performance compared to SOTA methods.

**Limitations And Societal Impact:**

Yes.

**Main Review:**

### Novelty and Significance
- The problem addressed is very important in many domains. Both the contributions regarding the introduction of series decomposition in the deep learning model and the auto-correlation mechanism are novel.

### Quality
- The auto-correlation mechanism achieves better performance than the self-attention mechanism and its several variants in terms of both speed and accuracy. While the self-attention mechanism is fairly explanatory,  It is not clear what the weighted sum of delayed sequences in equation 6 represents. The authors should provide intuitive explanations why auto-correlation works better than self-attention in the case of time series.
- The autocorrelation of a univariate time series tells us about the similarity between the original time series and its delayed version. But in the case of multivariate time series, different dimensions can have different time periods. How do the authors compute the autocorrelation in that case? Is it computed individually per dimension?
- The encoder inputs as noted in line 127 is of length I and hence the output of the encoder should also be of length I. How do authors resize the encoder output to length O in line 179? What is the rationality behind using only the second half of the seasonality and trend-cycle component as an input to the decoder?
- How do authors perform autocorrelation mechanism on the key and query sequences of variable length? This is the case when the output of both encoder and decoder is fed to the second auto-correlation block in decoder?
- Since the experiments have been repeated three times, the authors should also report the standard deviation in the MAE/MSE scores.

### Clarity
- The paper is well written and easy to follow.
- line 83 - have shown great power in modeling
- line 146 decoder - decoding

**Time Spent Reviewing:**

5

---

> ### Author Response · Authors · 2021-08-08
> **Response to Reviewer GKTE**
>
> Many Thanks to Reviewer GKTE for providing the thorough insightful comments.
>
> **Q1:** The meaning for weighted sum of delayed sequences in Equation 6.
>
> The lags of delayed sequences are selected based on the calculated autocorrelations that represent the estimated periods. So the delay operation is to align the sub-processes among periods. Since a higher auto-correlation indicates a more similar lagged process, the weighted sum adaptively aggregates the information from similar processes ($\underline{\text{Figure 3(d) of main paper}}$).
>
> Corresponding to the self-attention, which is a weighted sum of points, the proposed weighted sum of sequences provides series-wise aggregation and is more efficient for utilizing relevant information.
>
>
>
> **Q2:** Why Auto-Correlation performs better than self-attention? More intuitive explanations.
>
> First, more information does not mean better performance. Full attention with dense connections may include some irrelevant points and result in distraction. The sparse connection may lose some information but focus on the relevant points. Thus, as shown in $\underline{\text{Table 3 of main paper}}$, several sparse versions can also outperform full attention in many cases.
>
> Second, Auto-Correlation performs better than the self-attention family in the following two aspects:
>
> (1) **Discover the relevant information more sufficiently and precisely.** As shown in $\underline{\text{Figure 5 of main paper}}$, we visualize the learned dependencies from Auto-Correlation and various attentions. Auto-Correlation can capture the right sub-processes without omissions or errors. This means that without full connections, Auto-Correlation can utilize the series information even better.
>
> (2) **Aggregate the relevant information underlying the context of time series.** As shown in the intuitive comparision in $\underline{\text{Figure 3(d) of main paper}}$​, Auto-Correlation aggregates the relevant information from sub-series  to sub-series. While, attentions aggregate the information point by point, regardless of the time series context.
>
>
>
> **Q3:** Auto-Correlation for multivariate time series where different dimensions have different periods.
>
> (1) As stated in $\underline{\text{Section 8.1 of supplementary material}}$, take the ETT dataset as an example. It contains 7 series. We input this 7-dimension multivariate series to Autoformer after the linear-layer embedding, which fuses the raw 7 series and gets the deep representation with 512 channels. Then we **calculate the autocorrelation for the deep representation along the temporal dimension**.
>
> Also, we can treat the multivariate series separately per dimenstion. Here are the comparisons. At inference, Autoformer is faster than "separately per dimension" by a magnitude of the number of dimensions.
>
> | Input-96-predict-336 (ETT dataset)                   | MSE       | MAE       |
> | ---------------------------------------------------- | --------- | --------- |
> | Autoformer                                           | **0.351** | **0.384** |
> | Autoformer (separately per dimenstion of raw series) | 0.362     | 0.436     |
>
> (2) The concern that different series have different periods is reasonable. As per your request, to verify Autoformer's performance in this situation, we mix the ETT data (collected every 15 minutes) and the resampled ETT data (collected every 1 hour) along the input dimension, regardless of the mismatch of time stamps. This mixed dataset will contain two different periods at different dimensions of each time point. Autoformer still works well.
>
> | **Input-96-predict-336 (mixed dataset)**             | **MSE**   | **MAE**   |
> | ---------------------------------------------------- | --------- | --------- |
> | Autoformer                                           | **0.321** | **0.441** |
> | Autoformer (separately per dimenstion of raw series) | 0.329     | 0.449     |
> | Transformer                                          | 1.249     | 0.865     |
>
> We will include these ablations into the supplementary material.
>
>
>
> **Q4:** The details of the encoder's input and output length & the key and query sequences of variable length.
>
> The details about "resize" have been provided in $\underline{\text{lines 93-96 of supplemental material}}$. For variable lengths, we truncate the longer ones or fill the shorter ones by zeros. This makes the tensor of the keys and values of the same length as the queries.
>
>
>
> **Q5:** The input length of the decoder.
>
> For the fairness of comparison, we follow the decoder input length (I/2+O) in Informer [40]. As per your request, we perform the results with decoder input as I+O. It also works well but with higher computational and memory cost. We will include this ablation into the supplementary material.
>
> | Input-96-predict-336 (ETT dataset) | MSE   | MAE   |
> | ---------------------------------- | ----- | ----- |
> | Autoformer decoder input I/2+O     | 0.351 | 0.384 |
> | Autoformer decoder input I+O       | 0.349 | 0.382 |
>
>
>
> **Q6:** Standard deviation.
>
> Due to space limitation of the main paper, we have provided the standard deviation in $\underline{\text{Table 3 of supplemental material}}$.

---

> > ### Comment · Reviewer_GKTE · 2021-08-30
> > **Response**
> >
> > Thanks for addressing my concerns and providing additional experiments.

---

### Official Review · Reviewer_zw1M · 2021-07-29

**Rating:** 6
**Confidence:** 4

**Summary:**

This paper presents Autoformer to perform long-term time series forecasting. The key idea is to leverage an auto-correlation mechanism to discover the sub-series similarity based on the series periodicity and aggregate similar sub-series from underlying periods. The experiment results on several datasets showed the effectiveness of the proposed method.

**Ethics Review Area:**

["I don’t know"]

**Limitations And Societal Impact:**

Yes.

**Main Review:**

Strengths:
* This paper is well written and organized.
* Long term time series forecasting is an interesting problem to investigate.
* The proposed auto-correlation mechanism is innovative and technically sound.

Weaknesses:
* The assumption of Autoformer is not clear.
* The setup of long term forecasting needs to be justified.
* Several related works are not mentioned or compared.

Overall, although the idea of calculating the autocorrelation over frequency domain is not new (e.g., Vlachos et al., SDM 2005), the way to combine it with a multi-head attention mechanism indeed exhibits novelty to reduce the computational cost. The following are several concerns over this work.

It is not clear under what assumption/condition Autoformer can work well. The authors mentioned that the auto-correlation mechanism is developed based on series periodicity. For time series which do not exhibit clear periodicity, whether it still works or not? For example, for the exchange dataset, whether the long term prediction here is really meaningful or useful?

In the experiment, the input sequence length is 24/96, while the prediction lengths are 92, 192, … 720. I am wondering whether the input sequence can provide sufficient info to give a legitimate prediction. For electricity, traffic, and weather, this could be true, but I am not sure this is the case for exchange or other time series. Is there a theoretical justification that under what condition we can obtain a  legitimate prediction?

ARMA usually works well to capture the seasonal info but is not compared in the experiment. Furthermore, several related works are not compared

[1] Think Globally, Act Locally: A Deep Neural Network Approach to High-Dimensional Time Series Forecasting, NIPS 2019
[2] N-BEATS: NEURAL BASIS EXPANSION ANALYSIS FOR INTERPRETABLE TIME SERIES FORECASTING, ICLR 2020

Whether Autoformer works for multivariate time series?

How to determine tau in practice?

I could be persuaded if the above questions are clarified.


**Time Spent Reviewing:**

3 hours

---

> ### Author Response · Authors · 2021-08-08
> **Response to Reviewer zw1M**
>
> We would like to sincerely thank Reviewer zw1M for providing the insightful review and valuable comments.
>
> **Q1:** For time series without clear periodicity, whether Auto-Correlation still works or not?
>
> In $\underline{\text{Table 3 of main paper}}$, we have shown that Auto-Correlation outperforms the self-attention family on the ETT dataset (with clear periodicity), by replacing the Auto-Correlation in Autoformer with different self-attention modules. As per your suggestion, we repeat the experiment on the Exchange dataset (without clear periodicity). Here are more results:
>
> | Exchange (without clear periodicity) input-96-predict-336    | MSE       | MAE       |
> | ------------------------------------------------------------ | --------- | --------- |
> | Auto-Correlation + Autoformer architecture (deep decomposition) | **0.488** | **0.510** |
> | Full Self-Attention + Autoformer architecture (deep decomposition) | 0.632     | 0.584     |
> | LogSparse Attention + Autoformer architecture (deep decomposition) | 0.569     | 0.592     |
> | LSH Attention + Autoformer architecture (deep decomposition) | 0.553     | 0.549     |
> | PropSparse Attention + Autoformer architecture (deep decomposition) | 0.958     | 0.729     |
>
> As shown in the above table, for series without clear periodicity, Auto-Correlation still outperforms other self-attention mechnisms in both MSE and MAE. Therefore, Auto-correlation is a robust deep learning module for general-pattern series data.
>
> We have also provided some visualization results on the Exchange dataset (without clear periodicity) in $\underline{\text{Section 4.2 of supplementary material}}$, which shows that Autoformer can make meaningful long-term forecasting for series without clear periodicity. It is notable that both the Auto-Correlation and the decomposition architecure contribute substantially to Autoformer's predicative power for complex series.
>
>
>
> **Q2:** Whether the input sequence can provide sufficient information or not?
>
> In the paper, we follow standard evaluation protocol of LogTrans [16], which uses **fixed input length for different forecasting horizons**. We further extend the forecasting horizon to 720 for a fair comparison with Informer [40].
>
> We agree with your opinion that, for the electricity, traffic, weather datasets, setting the input length as 96 is enough to provide sufficient information. For the exchange and ILI datasets, as per your request, we further perform experiment which confirms that, a longer input sequence does provide more information and promote Autoformer's performance.
>
> | Autoformer (MSE \| MAE) | Input-96       | Input-192      | Input-336      | Input-720      |
> | ----------------------- | -------------- | -------------- | -------------- | -------------- |
> | Exchange (predict 96)   | 0.134 \| 0.270 | 0.120 \| 0.259 | 0.114 \| 0.244 | 0.094 \| 0.225 |
> | Exchange (predict 192)  | 0.272 \| 0.374 | 0.210 \| 0.329 | 0.186 \| 0.314 | 0.184 \| 0.308 |
> | Exchange (predict 336)  | 0.488 \| 0.510 | 0.420 \| 0.472 | 0.404 \| 0.460 | 0.392 \| 0.459 |
> | Exchange (predict 720)  | 1.367 \| 0.901 | 1.212 \| 0.844 | 1.147 \| 0.838 | 1.142 \| 0.831 |
>
> | Autoformer (MSE \| MAE) | Input-24       | Input-36       | Input-48       | Input-60       |
> | ----------------------- | -------------- | -------------- | -------------- | -------------- |
> | ILI (predict 24)        | 3.825 \| 1.345 | 3.055 \| 1.189 | 2.738 \| 1.090 | 2.655 \| 1.040 |
> | ILI (predict 36)        | 3.319 \| 1.216 | 2.674 \| 1.024 | 2.544 \| 0.998 | 2.344 \| 0.995 |
> | ILI (predict 48)        | 2.854 \| 1.122 | 2.749 \| 1.095 | 2.717 \| 1.082 | 2.715 \| 1.075 |
> | ILI (predict 60)        | 3.227 \| 1.232 | 3.077 \| 1.219 | 2.822 \| 1.103 | 2.742 \| 1.014 |
>
>
>
> **Q3:** More baselines for comparison, including ARIMA, DeepGLO, and N-BEATS.
>
> We have provided in $\underline{\text{Table 4 of supplementary material}}$​ the results of ARMIA, as well as the well-known deep learning model (DeepAR) and a strong statistical model (Prophet, open-sourced by FaceBook). Here are more results of Autoformer comparing with ARIMA, DeepGLO, and N-BEATS.
>
> (1) Results of input-96-predict-O under the **univariate** setting.
>
> | ETT (MSE \| MAE) | Predict 96         | Predict 192        | Predict 336        | Predict 720        |
> | ---------------- | ------------------ | ------------------ | ------------------ | ------------------ |
> | ARIMA            | 0.568 \| 0.572     | 0.804 \| 0.720     | 1.438 \| 1.010     | 3.291 \| 1.569     |
> | DeepGLO          | 0.199 \| 0.341     | 0.223 \| 0.360     | 0.245 \| 0.400     | 0.328 \| 0.462     |
> | N-BEATS          | 0.257 \| 0.389     | 0.298 \| 0.424     | 0.320 \| 0.445     | 0.363 \| 0.480     |
> | Autoformer       | **0.065 \| 0.189** | **0.110 \| 0.258** | **0.145 \| 0.295** | **0.182 \| 0.335** |
>
> | Exchange (MSE \|MAE) | Predict 96         | Predict 192        | Predict 336        | Predict 720        |
> | -------------------- | ------------------ | ------------------ | ------------------ | ------------------ |
> | ARIMA                | 0.308 ｜ 0.396     | 1.305 ｜ 1.178     | 1.762｜1.445       | 5.017 ｜ 1.893     |
> | DeepGLO              | 0.850 \| 0.786     | 1.825 \| 1.185     | 2.210 \| 1.330     | 5.818 \| 2.232     |
> | N-BEATS              | 0.319 \| 0.433     | 0.706 \| 0.651     | 1.282 \| 0.879     | 2.757 ｜1.341      |
> | Autoformer           | **0.126 \| 0.268** | **0.530 \| 0.565** | **0.586 \| 0.572** | **1.838 \| 1.201** |
>
> (2) Results of input-96-predict-O under the **multivariate** setting. For ARIMA, N-BEATS (univariate predictive model), we predict the multivariate series dimension by dimension.
>
> | ETT (MSE \|MAE) | Predict 96         | Predict 192        | Predict 336        | Predict 720        |
> | --------------- | ------------------ | ------------------ | ------------------ | ------------------ |
> | ARIMA           | 0.267 \| 0.382     | 2.414 \| 0.588     | 10.083 \| 0.896    | 15.338 \| 1.183    |
> | DeepGLO         | 0.288 \| 0.395     | 0.510 \| 0.551     | 0.872 \| 0.734     | 2.173 \| 1.208     |
> | N-BEATS         | 0.313 \| 0.395     | 0.392 \| 0.440     | 0.464 \| 0.473     | 0.571 \| 0.521     |
> | Autoformer      | **0.194 \| 0.284** | **0.261 \| 0.323** | **0.351 \| 0.384** | **0.491 \| 0.470** |
>
> | Exchange (MSE \|MAE) | Predict 96         | Predict 192        | Predict 336        | Predict 720        |
> | -------------------- | ------------------ | ------------------ | ------------------ | ------------------ |
> | ARIMA                | 0.327｜ 0.417      | 0.656  \|  0.568   | 0.970 \| 0.572     | 4.808 \| 1.182     |
> | DeepGLO              | 0.928 \| 0.751     | 1.142 \| 0.861     | 1.512 \| 1.013     | 1.542 \| 1.097     |
> | N-BEATS              | 0.316 \| 0.409     | 0.328 \| 0.444     | 1.203 \| 0.819     | 1.672 \| 1.013     |
> | Autoformer           | **0.134 \| 0.270** | **0.272 \| 0.374** | **0.488 \| 0.510** | **1.367 \| 0.901** |
>
> We will make $\underline{\text{Table 1 of main paper}}$​ more complete by adding the above results for all datasets and forecasting horizons.
>
>
>
> **Q4:** Whether Autoformer works for multivariate time series?
>
> We have implemented the multivariate experiments in $\underline{\text{Table 1 of main paper}}$ and univariate experiments in $\underline{\text{Table 4 of supplemental material}}$. The results demonstrate that Autoformer works well for both multivariate and univariate time series.
>
>
>
> **Q5:** The $\tau$ determination.
>
> The $\tau$'s represent the lag for series, and each $\tau$ yields an Auto-Correlation value (the similarity between the original series and its lagged version). Hence the $\tau$'s are selected based on the top-k values of calculated Auto-Correlations ($\underline{\text{Equation 6 of main paper}}$).

---

> > ### Author Response · Authors · 2021-08-18
> > **Reviewer feedback to author response**
> >
> > Dear Reviewer zw1M,
> >
> > Many thanks for your time and efforts in reviewing our paper.
> >
> > We kindly remind that we are less than one week into the discussion period. We have made extensive effort to try to successfully address your concerns and answer your questions, by providing all supporting experiments you requested and highlighting some results originally presented in the $\underline{\text{main paper}}$​ and the $\underline{\text{supplementary material}}$​.
> >
> > If you have any further concerns or questions, please do not hesitate to let us know, and we will respond to them timely.
> >
> > All the best,
> > Authors

---

> > > ### Comment · Reviewer_zw1M · 2021-08-31
> > > **Thanks for the response**
> > >
> > > I would like to thank the authors for responding to my previous questions. I read the paper and responses carefully.  Q3, Q4, and Q5 are clear to me right now.
> > >
> > > For Q1,  only one dataset (the exchange dataset) is used for empirical study as a non-periodical case. I wonder whether this can generalize to other non-periodical cases. In addition, for the exchange dataset, I am concerned about whether the current reported long-term MAE/MSE/results could provide practical guidance for real-world applications.
> > >
> > > For Q2,  I wonder whether there is a principled way (theory or empirical) to determine the best input length and prediction horizon? Especially for non-periodical data.
> > >
> > > Finally, the potential limitations of Autoformer should be discussed.

---

> > > > ### Author Response · Authors · 2021-08-31
> > > > **Response to Reviewer zw1M**
> > > >
> > > > Many thanks to Reviewer zw1M for the response and constructive suggestions.
> > > >
> > > > **Q1-Part1:** Whether the Auto-Correlation can generalize to other non-periodical cases?
> > > >
> > > > Following your preliminary review, we have provided additional results on the **Exchange dataset** in our previous response. As per your request, here we provide further results on two more datasets for non-periodical situations.
> > > >
> > > > (1) **ILI dataset.** This records the influenza-like-illness data weekly, which is also without clear periodicity. In addition to the results in $\underline{\text{Table 1 of main paper}}$, we also provide more ablations as follows:
> > > >
> > > > | ILI (without clear periodicity) input-24-predict-48          | MSE       | MAE       |
> > > > | ------------------------------------------------------------ | --------- | --------- |
> > > > | Auto-Correlation + Autoformer architecture (deep decomposition) | **2.854** | **1.122** |
> > > > | Full Self-Attention + Autoformer architecture (deep decomposition) | 3.256     | 1.146     |
> > > > | LogSparse Attention + Autoformer architecture (deep decomposition) | 3.301     | 1.208     |
> > > > | LSH Attention + Autoformer architecture (deep decomposition) | 3.112     | 1.187     |
> > > > | PropSparse Attention + Autoformer architecture (deep decomposition) | 3.138     | 1.201     |
> > > >
> > > > (2) **COVID-19 dataset.** We provided a case study of daily recorded COVID-19 data in $\underline{\text{Section 6 of supplementary material}}$, where Autoformer can still outperform all other baselines. Note that this dataset is also without clear periodicity. For the detailed comparison, we repeat the experiments as follows:
> > > >
> > > > | COVID-19 Country 1 (without clear periodicity) input-7-predict-30 | MSE       | MAE       |
> > > > | ------------------------------------------------------------ | --------- | --------- |
> > > > | Auto-Correlation + Autoformer architecture (deep decomposition) | **0.261** | **0.319** |
> > > > | Full Self-Attention + Autoformer architecture (deep decomposition) | 0.273     | 0.338     |
> > > > | LogSparse Attention + Autoformer architecture (deep decomposition) | 0.318     | 0.391     |
> > > > | LSH Attention + Autoformer architecture (deep decomposition) | 0.309     | 0.373     |
> > > > | PropSparse Attention + Autoformer architecture (deep decomposition) | 0.360     | 0.454     |
> > > >
> > > > The above results show that Auto-Correlation still outperforms other self-attention mechanisms for series without clear periodicity, though the improvements are relatively smaller than the more periodic cases, which can be expected due to the property of Auto-Correlation. **These new results from the three datasets are good evidence for the generality of our model in non-periodical cases.**  It is notable that the Auto-Correlation and the decomposition architecture work collaboratively to improve Autoformer's predictive power for complex series. We will include these results in the revision to complete our experiments.
> > > >
> > > >
> > > >
> > > > **Q1-Part2:** Can long-term MAE/MSE results provide practical guidance for real-world applications?
> > > >
> > > > (1) As shown in $\underline{\text{Table 1 of main paper}}$, Autoformer achieves the state-of-the-art in both MSE and MAE metrics. These two metrics are widely used in real-world applications to evaluate the forecasting model performance, such as the well-established time-series toolkit $\texttt{Kats}$ of Facebook.
> > > >
> > > > (2) **Showcases of the Exchange dataset and COVID-19 dataset are meaningful.** As shown in $\underline{\text{Figures 6 and 8 of supplementary material}}$, we have provided some showcases of the Exchange dataset and COVID-19 dataset for the intuitive comparison. These cases show that **Autoformer can forecast the long-term trend, peaks and troughs reasonably well**. These results can deliver valuable insights for the series-change judgment. We believe this can be helpful for financial planning or epidemic prevention.
> > > >
> > > > In summary, Autoformer can **outperform the other baselines, and provide meaningful forecasting results with more insights into trend, period, and temporal dynamics**.
> > > >
> > > >
> > > >
> > > > **Q2:** How to determine the best input length and prediction horizon? Especially for non-periodical data.
> > > >
> > > > This is a very good point for the practical use of Autoformer. Based on our experimentation, we have the following suggestions for setting these hyperparameters.
> > > >
> > > > (1) The prediction horizon is usually decided based on the application's requirement. E.g., if the weather forecaster is to predict 72 hours (3 days) into the future, then the prediction horizon is set as this length.
> > > >
> > > > (2) For the input length, as shown in the previous results, the relation between input length and model performance is dataset-specific.
> > > > - For the dataset with clear periodicity, we can choose an input length that is larger than the period, such as the ETT dataset and Traffic dataset.
> > > > - For the dataset without clear periodicity, enlarging the input length usually can obtain better results, such as the Exchange and ILI datasets. However, as shown below, we have to trade-off GPU memory cost and model performance, which is highly relevant in deep learning applications. In practice, we can enlarge the input length within the GPU limitation.
> > > >
> > > > | GPU memory cost (Training phase)         | Autoformer | Transformer |
> > > > | ---------------------------------------- | ---------- | ----------- |
> > > > | Input-96-predict-720 (Exchange dataset)  | 4564 MB    | 5356 MB     |
> > > > | Input-336-predict-720 (Exchange dataset) | 6332 MB    | 9912 MB     |
> > > > | Input-720-predict-720 (Exchange dataset) | 8466 MB    | 16926 MB    |
> > > >
> > > > In general, the input length determination is an experimental conclusion and has to adjust based on the data patterns. Note that this is a **common situation shared by well-established deep models** (Transformers, Informer, Reformer, etc.). Also, for the classic method ARIMA, $p$ is the order (number of time lags) of the autoregressive part of ARIMA model and also has to adjust following the data patterns. Some well-known ARIMA parameter auto-searching algorithms **also use a brute-force method** to optimize a certain criterion, such as the "Akaike Information Criterion".
> > > >
> > > >
> > > >
> > > > **Q3:** Limitations of Autoformer.
> > > >
> > > > We mostly employ the fixed input length for a clear comparison among different forecasting horizons. The input length $I$ is dataset-specific in real-world applications. And the best input length needs to be adjusted according to the data patterns and GPU memory limitation of devices.

---

> > > > > ### Comment · Reviewer_zw1M · 2021-09-01
> > > > > **Thanks for the response**
> > > > >
> > > > > Thanks for addressing my remaining concerns. I am pleased with the authors' responses and would like to raise my score from 5 to 6 providing that the authors can incorporate related updates in the manuscript.

---

### Author Response · Authors · 2021-08-28
**Summary**

We sincerely thank all the reviewers for their insightful reviews and valuable comments, which are instructive for us to further improve our paper.

This paper extends the signal processing and stochastic process theories to design a completely different Autoformer model for long time-series forecasting. The decomposition in the signal processing theory is renovated into a deep decomposition block, while the Auto-Correlation in the stochastic process theory is renovated into an Attention-style block, both can be layered freely to grow a big capacity deep model. Autoformer is a new deep model in parallel to Transformer, and achieves the remarkable **38% relative promotion** on six practical datasets.

The reviewers generally held positive opinions of our paper, in that the long-term forecasting problem we addressed is “very important in many domains”, this paper provides “two novel architecture changes”, the proposed Auto-Correlation mechanism is “innovative and technically sound”, “the method is evaluated on six benchmark datasets” and “achieves better performance compared to SOTA methods”, and “the paper is well written and easy to follow”.

The reviewers also raised insightful and constructive concerns. We made every effort to address all the concerns by providing sufficient evidence and requested results. Here is the summary.

- We further investigated the performance of the Auto-Correlation mechanism under various conditions. (1) For time series without clear periodicity, as asked by **Reviewers zw1M and zbiv**, we first recalled the analysis in $\underline{\text{Section 4.2 of supplementary materials}}$. Also, we showed that Auto-Correlation can outperform all other Attention-based mechanisms on the Exchange dataset. (2) For multivariate time series where different dimensions have different periods, as mentioned by **Reviewer GKTE**, we provided the experiments on the ETT dataset and the combined dataset that matches the case exactly. Our method can beat the baseline that forecasts each dimension separately on both benchmarks and is more efficient. (3) For time series with complex seasonalities, as pointed out by **Reviewer jDpZ**, we visualized the learned lags on the Exchange dataset. The learned lags can reflect the monthly and quarterly seasonalities well. Also, we added two different datasets to create more complex seasonalities, in which Autoformer can still surpass all other baselines.

- **Reviewers zw1M, jDpZ, and zbiv** asked about the model performance of different input lengths. We showed that the relation between performance and input length is dataset-specific. For the ETT dataset with clear periodicity, relatively short inputs can provide sufficient information for long-term forecasting. But for the Exchange dataset that is more complex, enlarging the inputs will yield better results.

- **Reviewers GKTE and zbiv** had asked for more intuitive explanations of why Autoformer is better over other strong baselines. We first recalled the visualization in $\underline{\text{Figure 4 of main paper }}$ to show that the proposed decomposition can get more predictable components. Besides, based on the intuitive comparison and visualization in the main paper, we further clarify that Auto-Correlation can discover the relevant information more sufficiently and precisely and aggregate the relevant information underlying the context of time series.

- **Reviewer zw1M** had suggestions about the compared baselines. We further compared our model with DeepGLO and N-BEATS that the reviewer requested under both univariate and multivariate settings. Autoformer can also provide the best performance.

- **Reviewer jDpZ** had questions about the model ability for different time-scale patterns. We further clarify the time-scale that the model captured should consider both the input past and the forecasted future. Besides, the six benchmarks in our paper can cover the weekly, monthly, quarterly, and even longer patterns.

It is our great pleasure that **Reviewer jDpZ** has acknowledged our response. We'd be very happy to answer any further questions from **Reviewers zw1M, GKTE, zbiv**.

---

### Public Comment · ~HeeHun_Jeong1 · 2022-10-17
**Page 19 Algorithm 1 line 18. Dimension comment.**

From page 19 I found some trivial typo. Algorithm1 line 18 : They say the dimension of Xpred is R(I/2+O)*dmodel

But from my reasoning, Final output prediction should have dimension as follows, R(I/2+O)*d

Please check above statement and reply on this comment.

Thank you.

---

> ### Public Comment · Authors · 2022-10-17
> **Thanks for your comments.**
>
> Thanks a lot for your detailed comments.
>
> We will rephrase this typo as soon as possible.

---

### Decision · Program_Chairs · 2021-09-28

**Decision:**

Accept (Poster)

**Comment:**

This paper investigates the problem of long-term forecasting for time series models. It proposes a transformer-based architecture with an attention mechanism based on auto-correlation and combines it with a periodicity-based time series decomposition approach. The problem studied is relevant and important. The proposed architecture is novel although the basic concepts it builds on are well known in time series analysis. The proposed approach is technically sound and the empirical evaluation of the approach provides adequate evidence that it has the potential to provide improved performance on the long term time series forecasting problem relative to existing approaches. Following the author response and discussion, most initial reviewer concerns were addressed and the consensus is that the paper should be accepted. One important point for the authors to clarify are the limitations of the approach with respect to time series with different structural properties (e.g, complex seasonality, quasiperiodicity, absence of periodicity, etc.).

**Consistency Experiment:**

NeurIPS has a long history of experimentation. In 2014, NeurIPS ran an experiment in which 10% of submissions were reviewed by two independent committees to quantify the randomness in the review process. This year, we repeated a variant of this experiment to see how the quality of the review process has changed over time.  This paper was part of the experiment and was therefore assigned to two committees (consisting of reviewers, an Area Chair, and a Senior Area Chair) that reached independent decisions.  If both committees made the same recommendation, this recommendation was followed. If a single committee recommended acceptance, the paper was accepted (with the exception of a few cases in which the other committee identified what we considered a fatal flaw, e.g., an error in a key result).

Both committees reached the same decision: **Accept (Poster)**

The other committee assigned to the paper recommended **Accept (Poster)**.  You can find the other set of reviews, along with any follow up discussion with the authors here:
https://openreview.net/forum?id=I55UqU-M11y